# Rank-1 Approximation of Inverse Fisher
# for Natural Policy Gradients in Deep Reinforcement Learning

**Yingxiao Huo**[*]                              *yingxiao.huo@postgrad.manchester.ac.uk*
*Department of Computer Science*
*The University of Manchester*

**Satya Prakash Dash**[*]                        *satyaprakash.dash@postgrad.manchester.ac.uk*
*Department of Computer Science*
*The University of Manchester*

**Radu Stoican**[*]                              *radu.stoican@manchester.ac.uk*
*Department of Computer Science*
*The University of Manchester*

**Samuel Kaski**                                 *samuel.kaski@manchester.ac.uk*
*Department of Computer Science*
*The University of Manchester*
*Aalto University*

**Mingfei Sun**[†]                               *mingfei.sun@manchester.ac.uk*
*Department of Computer Science*
*The University of Manchester*

**Reviewed on OpenReview:** *https://openreview.net/forum?id=ko8Kn7TS6m*

[*]Equal contribution; [†]Corresponding author

## Abstract

Natural gradients have long been studied in deep reinforcement learning due to their fast convergence properties and covariant weight updates. However, computing natural gradients requires inversion of the Fisher Information Matrix (FIM) at each iteration, which is computationally prohibitive in nature. In this paper, we present an efficient and scalable natural policy optimization technique that leverages a rank-1 approximation to full inverse-FIM. We theoretically show that under certain conditions, a rank-1 approximation to inverse-FIM converges faster than policy gradients and, under some conditions, enjoys the same sample complexity as stochastic policy gradient methods. We benchmark our method on a diverse set of environments and show that it achieves superior performance to standard actor-critic and trust-region baselines.

## 1 Introduction

Policy gradient methods (Sutton et al., 1999) have emerged as one of the most prominent candidates for modern reinforcement learning (RL). Specifically, they have seen widespread adoption in deep RL, where they incorporate differentiable policy parametrization and function approximators to help policy gradient methods scale up and learn robust controllers for high-dimensional state spaces (Arulkumaran et al., 2017; Wang et al., 2022). Despite this, a fundamental limitation of policy gradients is that RL objectives tend to be highly non-concave in nature. Consequently, algorithms based on standard gradient descent, such as REINFORCE (Williams, 1992), are prone to getting stuck at sub-optimal local minima when training high-dimensional policies. This is due to the gradient giving only the direction in which a policy's parameters

should be optimized, without indicating the size of the optimization step (Martens, 2020). Step size selection is, therefore, non-trivial. Small steps lead to stable but slow and sample-inefficient training. On the contrary, large steps can cause unstable updates. This is particularly dangerous in RL, as unstable policies collect poor data, making it difficult for the agent to recover and escape local optima (van Heeswijk, 2022).

One approach to reduce this sub-optimality and achieve stable training with large steps is to incorporate the geometry of the loss manifold into account and thus consider natural gradients (Amari, 1998). Specifically, when applied to policy gradient RL, where policies are modeled as probability distributions, the Fisher information matrix (FIM) is used to precondition the policy gradients, which gives rise to natural policy gradient (NPG) methods (Kakade, 2001; Bagnell & Schneider, 2003; Peters et al., 2005). NPG algorithms are of interest because they generally tend to converge faster and have superior performance compared to their standard policy gradient counterparts (Grondman et al., 2012). More recently, NPGs established the roots of some of the most popular RL algorithms used in practice, including Trust Region Policy Optimization (TRPO) (Schulman et al., 2015a), Proximal Policy Optimization (PPO) (Schulman et al., 2017). Furthermore, the study of NPG methods is also motivated by their success in real-world applications (Richter et al., 2006; Su et al., 2019).

One of the main drawbacks of NPG algorithms is the high cost of computing and inverting the FIM. Storage is also an important concern, as for a parameterized policy, the FIM is a square matrix with dimensions equal to the number of parameters. Naturally, this issue becomes especially limiting when policies are parameterized by deep neural networks. In practice, NPG algorithms usually avoid computing the inverse Fisher directly and instead rely on approximations. A simple option is to only consider the diagonal elements of the FIM (Becker & Lecun, 1989; LeCun et al., 2002), leading to efficient inversion and low storage requirements. A more sophisticated approach is K-FAC (Martens & Grosse, 2015), which uses Kronecker products to approximate the FIM efficiently. As an alternative to the true FIM, the empirical FIM can be computed from samples, which is especially convenient when the required underlying probability distributions are unknown (Schraudolph, 2002). Hessian-free methods give a less direct approach to using the FIM without ever explicitly computing or storing it (Martens, 2010). As an example, TRPO avoids computing the FIM, calculating the Fisher-vector product directly instead. Despite the extensive literature on the subject, choosing the right way of approximating natural gradients is usually not straightforward. This choice often involves a trade-off between computational efficiency and approximation accuracy.

The goal of this work is to bridge the theoretical convergence properties of NPG with a practical, fast, and computationally feasible approach to invert FIM for high-dimensional reinforcement learning tasks. In this work, we use the empirical FIM as an alternative to the FIM and incorporate the Sherman-Morrison formula for efficient inversions. We propose a rank-1 approximation to avoid explicit matrix operations, keeping the time complexity at $\mathcal{O}(d)$ for a policy parameterized by a vector $\theta$ of dimension $d$, without the need for inner loops like conjugate gradient, thus balancing computational cost and accuracy of curvature information.

The main contributions are listed below:

1. We provide a fast and efficient way to compute the inverse FIM, which can be computed in $\mathcal{O}(d)$ complexity rather than $\mathcal{O}(d^3)$ for naive inversion of FIM.

2. We theoretically show that under certain conditions, our NPG approximation using the empirical FIM and Sherman-Morrison formula enjoys global convergence when using log-linear function approximators.

3. We also prove that the error induced in the policy update due to this rank-1 approximation is bounded and reduces with the increase in iterations.

4. We propose a novel policy gradient algorithm using this update: Sherman-Morrison Actor Critic (**SMAC**). We show that SMAC achieves faster convergence in diverse OpenAI Gym tasks (classic control and MuJoCo environments (Todorov et al., 2012)), compared to vanilla policy gradient, Adam, and trust-region methods that use Conjugate Gradient solvers to approximate the direction of natural policy gradient.

## 2 Related Work

Gradient descent updates all the parameters of a model equally, meaning that weights are considered to be inside a Euclidean space. However, the loss function of the model actually induces a Riemannian manifold space. Therefore, the size of a gradient step does not reflect the true scale of change of the model's loss. Consequently, standard gradient descent methods are highly sensitive to the choice of step size. To avoid this issue, Amari first framed gradient-based learning on a Riemannian manifold, introducing the natural gradient descent algorithm, which uses the update direction $F(\theta)^{-1}\nabla J(\theta)$, with the FIM $F(\theta)$ and objective $J(\theta)$ (Amari, 1998). Here, the FIM is a covariant measure of the amount of information contained in the parameters (Amari, 1998). Kakade extended this idea to reinforcement learning and coined the natural policy gradient (NPG) algorithm (Kakade, 2001). However, the computational cost associated with calculating the FIM, especially in models with a large number of parameters, can be prohibitively expensive. Therefore, practical implementations of natural gradients rely on approximations instead. We thus summarize the related work on these approximations as follows.

**Kronecker-factored Approximate Curvature** Block-diagonal approximations to the natural gradient, such as K-FAC (Martens & Grosse, 2015) and subsequent refinements (Zhang et al., 2023; Ren & Goldfarb, 2021; Bahamou et al., 2023; Benzing, 2022), have been widely adopted due to their computational efficiency. Related variants have also been developed for reinforcement learning (Wu et al., 2017).

**Hessian-Free Methods** Hessian-free optimization (Martens, 2010), another method for computing natural gradients, has seen widespread use in reinforcement learning, particularly due to its superior performance in TRPO (Schulman et al., 2015a). This approach employs the conjugate gradient (CG) method to solve the associated linear system, thereby avoiding explicitly storing FIM, keeping memory usage comparable to a single forward–backward pass. However, CG requires several additional forward–backward passes, which reduces computational efficiency. In addition, due to the limited precision of CG estimation, practitioners usually perform an extra KL-based line search to adapt the step size (Schulman et al., 2015a).

**Empirical Fisher Information** The empirical Fisher (EF) estimates the Fisher information matrix by replacing the exact expectation over gradient outer products with the outer product of the gradients computed from individual samples (Yang et al., 2022). However, the empirical Fisher's simple estimate has been frequently criticized for deviating significantly from the true Fisher, and its limitations have been analyzed in many studies (Martens, 2020; Kunstner et al., 2019). Nevertheless, its very low computational cost has enabled a broad range of successful applications (Roux et al., 2007; Ren & Goldfarb, 2019; Wu et al., 2024). Compared with the extensive use of K-FAC and Hessian-free methods in RL, the EF approach remains under-explored and is seldom applied in this domain.

## 3 Preliminaries

**Markov decision process (MDP)** We consider an MDP $(\mathcal{S}, \mathcal{A}, P, R, \gamma)$ with a continuous or discrete state space $\mathcal{S}$, action space $\mathcal{A}$, transition kernel $P(s'|s,a)$, bounded reward function $r : S \times A \mapsto [-R, R]$, and discount factor $\gamma \in (0, 1)$. We also consider a stochastic policy $\pi_\theta(a|s)$ with parameters $\theta \in \mathbb{R}^d$ that induces a trajectory distribution $\pi_\theta(\tau_i)$ over $\tau_i = (s_0^i, a_0^i, s_1^i, a_1^i, \dots)$.

**Performance objective and policy gradient** Under the standard actor-critic framework, the RL objective is given by the expected discounted return of $\pi_\theta$:

$$J(\theta) \;=\; \mathbb{E}_{\tau \sim \pi_\theta} \left[ \sum_{t=0}^{\infty} \gamma^t \, r(s_t, a_t) \right]. \tag{1}$$

To maximize Eq. 1, the advantage actor-critic (A2C) (Mnih et al., 2016) algorithm uses an advantage function $A(s,a) = Q(s,a) - V(s)$, defined through the critic's state-value and action-value functions

$$V(s) = \mathbb{E}_{\tau \sim \pi_\theta} \left[ G_t \mid s_t = s \right], \qquad Q(s,a) = \mathbb{E}_{\tau \sim \pi_\theta} \left[ G_t \mid s_t = s, a_t = a \right],$$

where $G_t = \sum_{l=0}^{\infty} \gamma^l r(s_{t+l}, a_{t+l})$. Following the policy gradient theorem, the A2C gradient is then given by

$$\nabla_\theta J(\theta) = \mathbb{E}_{\tau \sim \pi_\theta} \Big[\sum_{t=0}^{\infty} \nabla_\theta \log \pi_\theta(a_t|s_t) \, A(s_t, a_t)\Big].$$

**Fisher information matrix and Natural Policy Gradient** For the exponential-family policy $\pi_\theta(\cdot \mid s)$, the natural Riemannian metric in parameter space can be expressed in terms of state visitation distribution induced by the same policy $\pi_\theta(\cdot \mid s)$ as

$$\boldsymbol{F}_s(\theta) = \mathbb{E}_{a \sim \pi_\theta(\cdot|s)}\big[\nabla_\theta \log \pi_\theta(a|s) \nabla_\theta \log \pi_\theta(a|s)^\top\big]. \tag{2}$$

The invariant FIM $\boldsymbol{F}_\rho(\theta)$ can be obtained by averaging over the state visitation distribution leading to the natural gradient update

$$\theta^{k+1} = \theta^k + \eta \boldsymbol{F}_\rho(\theta^k)^{-1} \nabla_\theta J(\theta^k),$$
$$\boldsymbol{F}_\rho(\theta^k) = \mathbb{E}_{s \sim d_\rho^{\pi_\theta}}[\boldsymbol{F}_s(\theta)],$$

where $\eta$ is the step size and $d_\rho^{\pi_\theta}$ represents the state visitation distribution under a policy $\pi_\theta$ and initial state distribution $\rho$:

$$d_\rho^{\pi_\theta}(s) := (1 - \gamma) \, \mathbb{E}_{s_0 \sim \rho} \sum_{t=0}^{\infty} \gamma^t \, \mathbb{P}(s_t = s \mid s_0, \pi_\theta) + \gamma \rho(s).$$

**Empirical Fisher information** Replacing the expectation in $\boldsymbol{F}(\theta)$ with a sample average over one single trajectory $\tau = \{(s_i, a_i)\}_{t=0}^{T-1}$ gives the empirical Fisher approximation

$$\hat{\boldsymbol{F}}(\theta) = \frac{1}{N} \sum_{i=1}^{N} \nabla_\theta \log \pi_\theta(a_i|s_i) \nabla_\theta \log \pi_\theta(a_i|s_i)^\top.$$

**Sherman-Morrison formula** The Sherman–Morrison formula efficiently computes the inverse of a matrix after a "rank-1 update" has been applied to it, provided that the inverse of the original matrix was already known (Sherman & Morrison, 1950). Suppose $\boldsymbol{X} \in \mathbb{R}^{n \times n}$ is an **invertible square matrix** and $\boldsymbol{u}, \boldsymbol{v} \in \mathbb{R}^n$ are **column vectors**. Then $\boldsymbol{X} + \boldsymbol{u}\boldsymbol{v}^\top$ is invertible **iff**

$$1 + \boldsymbol{v}^\top \boldsymbol{X}^{-1} \boldsymbol{u} \neq 0.$$

In this case, the Sherman-Morrison formula allows the computation of the inverse

$$(\boldsymbol{X} + \boldsymbol{u}\boldsymbol{v}^\top)^{-1} = \boldsymbol{X}^{-1} - \frac{\boldsymbol{X}^{-1} \boldsymbol{u}\boldsymbol{v}^\top \boldsymbol{X}^{-1}}{1 + \boldsymbol{v}^\top \boldsymbol{X}^{-1} \boldsymbol{u}}. \tag{3}$$

If $\boldsymbol{X}^{-1}$ is already known, this matrix inversion formula can bypass computing the inverse $(\boldsymbol{X} + \boldsymbol{u}\boldsymbol{v}^T)^{-1}$ directly, avoiding the cubic scaling with the dimension of $\boldsymbol{X}$. Instead, the update only requires three vector operations, which scale linearly with the dimension of $\boldsymbol{X}$. This update can use the full power of current GPU hardware platforms, since computing vector operations can be heavily parallelized. At the same time, it avoids the time-intensive matrix inversion processes. Methods such as Singular Value Decomposition (SVD), which decomposes a matrix into its singular values and vectors, and LU decomposition, which factors a matrix into a lower triangular matrix (L) and an upper triangular matrix (U), are much slower than the simple operations used in our approach.

For the rest of the paper, we abbreviate the actual FIM at $\theta^k$, $\boldsymbol{F}(\theta^k)$ as $\boldsymbol{F}^k$ and the estimate of a batch as $\hat{\boldsymbol{F}}^k$ for notational convenience.

**Sherman-Morrison for computing the inverse Fisher** The Sherman-Morrison formula can be used to incrementally build the inverse of the EF $(\hat{\boldsymbol{F}}^k)^{-1}$ at step $k$ by reusing the previous estimate $(\hat{\boldsymbol{F}}^{k-1})^{-1}$ (Singh & Alistarh, 2020):

$$
\begin{aligned}
(\hat{\boldsymbol{F}}^k)^{-1}g^k &= \left[(\hat{\boldsymbol{F}}^{k-1} + \nabla_\theta \log \pi_\theta(a|s)\nabla_\theta \log \pi_\theta(a|s)^\top)\right]^{-1} g^k \\
&= (\hat{\boldsymbol{F}}^{k-1})^{-1}g^k - \frac{(\hat{\boldsymbol{F}}^{k-1})^{-1}\nabla_\theta \log \pi_\theta(a|s)\nabla_\theta \log \pi_\theta(a|s)^\top(\hat{\boldsymbol{F}}^{k-1})^{-1}}{1 + \nabla_\theta \log \pi_\theta(a|s)^\top(\hat{\boldsymbol{F}}^{k-1})^{-1}\nabla_\theta \log \pi_\theta(a|s)}g^k.
\end{aligned}
$$

Other approaches compute a fresh EF at each iteration (Wu et al., 2024), which is used in the natural gradient

$$
\Delta\theta = \eta(\lambda\boldsymbol{I} + \nabla_\theta \log \pi_\theta(a|s)\nabla_\theta \log \pi_\theta(a|s)^\top)^{-1}\log \pi_\theta(a|s),
$$

where $\lambda > 0$ is a fixed damping coefficient.

## 4 NPG Approximation

We adopt a matrix-free scheme for approximating the FIM to reduce memory and computational cost. At each step, we form a local empirical Fisher from the current batch, add a damping term $\lambda$, and apply the Sherman–Morrison formula to compute the inverse–vector product $\boldsymbol{F}(\theta)^{-1}g^k$ directly, where $g^k$ is the policy gradient. This approach requires $\mathcal{O}(d)$ memory.

This formulation captures the local curvature of the loss landscape using only the current gradient direction. The damping term $\lambda$ ensures numerical stability and prevents the update direction from being overly sensitive to individual gradients. Compared to the exact FIM, this method strikes a balance between computational efficiency and curvature-informed updates.

### 4.1 Matrix-Free Update

The time and space complexity of explicitly computing and storing the FIM is $\mathcal{O}(d^2)$. So, maintaining a previous estimate $(\hat{\boldsymbol{F}}^{k-1})^{-1}$ and applying $(\hat{\boldsymbol{F}}^k)^{-1}g$ can be infeasible for large-scale models. In this work, we instead use the Sherman–Morrison formula to directly compute the product of the inverse empirical Fisher and the policy gradient.

We consider a one-sample empirical Fisher with damping, take $(s_k, a_k) \sim d^{\pi_{\theta^k}}(s)\pi_{\theta^k}(a \mid s)$, that is, one state-action pair visited when rolling out $\pi_{\theta^k}$:

$$
\hat{\boldsymbol{F}}^k = \lambda\boldsymbol{I} + \boldsymbol{F}^k(\theta) = \lambda\boldsymbol{I} + \nabla_\theta \log \pi_{\theta^k}(a_k|s_k)\nabla_\theta \log \pi_{\theta^k}(a_k|s_k)^\top.
$$

Applying the Sherman–Morrison formula then yields:

$$
\begin{aligned}
(\hat{\boldsymbol{F}}^k)^{-1} &= (\lambda\boldsymbol{I} + \nabla_\theta \log \pi_{\theta^k}(a^k|s^k)\nabla_\theta \log \pi_{\theta^k}(a^k|s^k)^\top)^{-1} \\
&= \frac{1}{\lambda}\boldsymbol{I} - \frac{\nabla_\theta \log \pi_{\theta^k}(a^k|s^k)\nabla_\theta \log \pi_{\theta^k}(a^k|s^k)^\top}{\lambda^2 + \lambda\nabla_\theta \log \pi_{\theta^k}(a^k|s^k)^\top\nabla_\theta \log \pi_{\theta^k}(a^k|s^k)} \\
&= \frac{1}{\lambda}\boldsymbol{I} - \frac{\boldsymbol{F}^k}{\lambda^2 + \lambda\operatorname{Tr}(\boldsymbol{F}^k)},
\end{aligned}
\tag{4}
$$

Multiplying the inverse Fisher by the gradient gives the natural policy gradient update direction

$$
\Delta\theta_k = \eta\,(\hat{\boldsymbol{F}}^k)^{-1}g^k = \eta\left[\frac{1}{\lambda}g^k - \frac{\nabla_\theta \log \pi_{\theta^k}(a^k|s^k)\nabla_\theta \log \pi_{\theta^k}(a^k|s^k)^\top g^k}{\lambda^2 + \lambda\nabla_\theta \log \pi_{\theta^k}(a^k|s^k)^\top \nabla_\theta \log \pi_{\theta^k}(a^k|s^k)}\right],
\tag{5}
$$

where $g^k = \nabla_\theta \log \pi_{\theta^k}(a^k|s^k)A(s,a)$ is the one-sample policy gradient update direction, $A(s,a)$ is the advantage estimate and $\eta$ is the step size.

This update involves only a matrix-vector operation and a vector outer product, making its time and memory complexity comparable to that of first-order methods, such as stochastic policy gradient.

### 4.2 Convergence Analysis

We take inspiration from the global convergence guarantees provided by Agarwal et al. (2021) and a follow-up work by Liu et al. (2022), which improved the global convergence results. To prove the global convergence, we take a standard Lipschitz policy assumption (Assumption 3) and average performance difference bounds (Lemma 3 from Liu et al. (2022)) and Lemma 2. We provide a theoretical analysis showing that the Sherman–Morrison rank-1 approximation to the inverse Fisher achieves global convergence. Under strong convexity assumptions (already taken in Kakade (2001); Peters et al. (2005)) of the FIM, we show that the average performance between the Sherman-Morrison update and the optimal policy $\pi^*$ can be bounded in terms of the the variance of policy gradient estimator ($\sigma^2$), discount factor ($\gamma$), the error in the Advantage estimate induced by the neural approximator ($\varepsilon_{bias}$) and the Lipschitz constant of the policy gradient ($L_J$). We defer the proof to the appendix, and state the final theorem here.

**Theorem 1.** *If we take the stochastic Sherman-Morrison policy update in Equation 4 and take $\eta = \frac{1}{4L_J}$, $K = \mathcal{O}\left(\frac{1}{(1-\gamma)^2\varepsilon^2}\right)$, $N = \mathcal{O}\left(\frac{\sigma^2}{\varepsilon^2}\right)$, and $H = \mathcal{O}\left(\log(\frac{1}{(1-\gamma)\varepsilon})\right)$. Then, we have*

$$J(\pi^\star) - \frac{1}{K}\sum_{k=0}^{K-1}\mathbb{E}[J(\theta^k)] \leq \frac{\sqrt{\varepsilon_{bias}}}{1-\gamma} + \varepsilon.$$

*In total, stochastic PG samples $\tilde{\mathcal{O}}\left(\frac{\sigma^2}{(1-\gamma)^2\varepsilon^4}\right)$ trajectories, where $\varepsilon_{bias}$ is the approximation error from the advantage function, and $\varepsilon$ is an arbitrary precision level and $\tilde{\mathcal{O}}$ hides all the logarithmic factors.*

Our proof relies on the fact that performance bound given by Lemma 3 (Liu et al., 2022) and we prove that this result holds with a sample complexity of $KN = \mathcal{O}(\frac{\sigma^2}{(1-\gamma)^2\varepsilon^4})$, where $N$ is the number of trajectories per update and $K$ is the number of updates. This shows that to get an $\varepsilon$ accurate policy, one needs $\mathcal{O}(\frac{1}{\varepsilon^2})$ samples, and when the horizon $H$ and number of samples $N$ are large enough, then $\sigma$ can be reduced to arbitrary precision. We would like to point out that the non-degeneracy of FIM [Assumption 2] and compatible function approximator [Assumption 4] are standard assumptions in stationary convergence of PG adapted by Agarwal et al. (2021) and can be realized by gaussian and softmax parametrized policies (See Section B.2 of Liu et al. (2022) and Section 8 of Ding et al. (2022) for further discussion). For compatible function approximation the value of $\varepsilon_{bias}$ can be arbitrarily small for neural approximators (Wang et al., 2019) and is exactly zero for softmax parametrized policies (Ding et al., 2022). Detailed derivations can be found in Appendix 2.

Note that, the obtained bound stems from average regret to the global optimum and recently Yuan et al. (2022) have obtained $\tilde{\mathcal{O}}(\varepsilon^{-3})$ bound on the condition when the approximator error is negligible, or $\varepsilon_{bias} = 0$, but with a finite $\varepsilon_{bias}$ the authors have recovered a bound of $\tilde{\mathcal{O}}(\varepsilon^{-4})$, similar to ours.

### 4.3 A Toy Problem

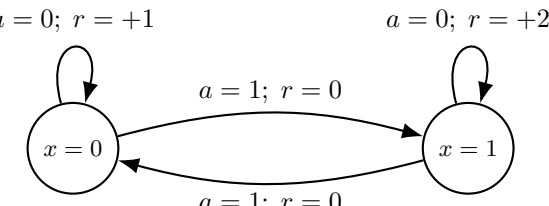

Figure 1: An example two-state MDP where each state $x$ has two control actions, one which self-transition with reward +1 or +2) and another control that transitions the initial state to the other state with zero reward).

We adopt a classical reinforcement–learning MDP problem, been used by Kakade (2001) & Bagnell & Schneider (2003) with two states $x \in \{0, 1\}$ and two discrete actions $a \in \{0, 1\}$. The policy is parameterized

by a vector $\boldsymbol{\theta} = [\theta_0, \theta_1]^\top$, where each component governs an independent Bernoulli distribution over actions. Let the probability of taking action $a = 0$ in state $x$ be

$$\pi_{\boldsymbol{\theta}}(a = 0 \mid x = 0) = \frac{1}{1 + e^{\lambda_p \theta_0}}, \tag{6}$$

$$\pi_{\boldsymbol{\theta}}(a = 0 \mid x = 1) = \frac{1}{1 + e^{\theta_1}} \tag{7}$$

where $\lambda_p$ scales the first policy parameter and is used to test scaling invariance. Assume that the agent starts at state $x = 0$, then the objective function of the the MDP can be expressed as

$$J(\boldsymbol{\theta}) = \pi_{\boldsymbol{\theta}}(a = 0 \mid x = 0) + 2(1 - \pi_{\boldsymbol{\theta}}(a = 0 \mid x = 0))\, \pi_{\boldsymbol{\theta}}(a = 0 \mid x = 1). \tag{8}$$

Note that, for a converged policy the agent shall transit to state $x = 1$ and remain there and the final return shall be $+2$ which is the the action of self transitioning at state $x = 1$. So we start from writing the analytic policy gradient vector,

$$\nabla_{\boldsymbol{\theta}} J(\boldsymbol{\theta}) = \begin{bmatrix} -\lambda_p\, \pi_0(1 - \pi_0)(1 - 2\pi_1) \\ -2(1 - \pi_0)\pi_1(1 - \pi_1) \end{bmatrix}. \tag{9}$$

and the FIM can be calculated as:

$$F(\boldsymbol{\theta}) = \begin{bmatrix} \lambda_p^2\, \pi_0(1 - \pi_0) & 0 \\ 0 & \pi_1(1 - \pi_1) \end{bmatrix}. \tag{10}$$

Then NPG update direction can be evaluated as

$$\dot{\boldsymbol{\theta}}_{\mathrm{NPG}} = F(\boldsymbol{\theta})^{-1}\, \nabla_{\boldsymbol{\theta}} J(\boldsymbol{\theta}) = \begin{bmatrix} \dfrac{-(1 - 2\pi_1)}{\lambda_p} \\ -2(1 - \pi_0) \end{bmatrix}. \tag{11}$$

This direction eliminates the dependence on the arbitrary scaling $\lambda_p$, making it invariant under smooth reparameterizations of $\boldsymbol{\theta}$ as shown by Bagnell & Schneider (2003).

Now the SM update direction in this setting can be evaluated as

$$\dot{\boldsymbol{\theta}}_{\mathrm{SM}} = \frac{1}{\lambda} \left( \boldsymbol{I} - \frac{F(\boldsymbol{\theta})}{\lambda + Tr(F(\boldsymbol{\theta}))} \right) \nabla_{\boldsymbol{\theta}} J(\boldsymbol{\theta}) \tag{12}$$

$$= \begin{bmatrix} \dfrac{1}{\lambda}(-\lambda_p\, \pi_0(1 - \pi_0)(1 - 2\pi_1)) - \dfrac{\lambda_p^3\, \pi_0^2(1 - \pi_0)^2(1 - 2\pi_1)}{\lambda^2 + \lambda\big[\lambda_p^2\, \pi_0(1 - \pi_0) + \pi_1(1 - \pi_1)\big]} \\ \dfrac{1}{\lambda}(-2(1 - \pi_0)\pi_1(1 - \pi_1)) - \dfrac{2\, \pi_1^2(1 - \pi_1)^2(1 - \pi_0)}{\lambda^2 + \lambda\big[\lambda_p^2\, \pi_0(1 - \pi_0) + \pi_1(1 - \pi_1)\big]} \end{bmatrix} \tag{13}$$

where $\lambda$ is the damping factor. This type of approximation although looks highly nonlinear but we can carefully tune $\lambda$ and make the update less sensitive to $\lambda_p$.

As shown in Fig.2, the plot in the left panel update the trajectories in the log-odds space for all damping settings $\lambda_p \in \{0.75, 1.0, 2.0\}$, Compared with the PG updates, the SM updates follow a direction much closer to that of NPG and maintain a step size more consistent. And PG updates deviate significantly as $\lambda$ changes, showing its sensitivity to parameter scaling. The plot in the right panel shows the returns over iterations, where SM update achieves faster and more stable improvement than PG and approaches the same final performance as NPG. Therefore, the SM update preserves the geometric characteristics of the NPG while being more computational efficient in high-dimensional cases.

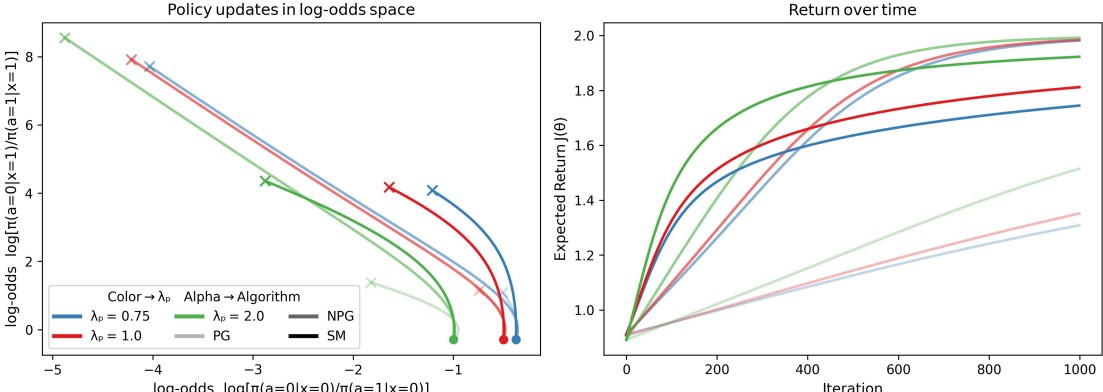

Figure 2: Results on toy MDP problem: Left side plot represents the log-odds plot for state $x = 0$ in x-axis and state $x = 1$ in y-axis and the right side plot shows the returns over iteration for PG, NPG and SM updates for different $\lambda_p \in \{0.75, 1.0, 2.0\}$ coefficient in policy parametrization. Note that, the least intense lines represent PG direction, medium is NPG and maximum deep lines represent SM update direction.

## 4.4 SM-ActorCritic

We plug our approximation into a standard A2C algorithm. Let $\pi_\theta$ be the policy (actor) and $V_\phi$ the value network (critic), with parameters $\theta$ and $\phi$, respectively. At each iteration, we collect $T$ transition tuples and compute the generalized advantage estimator (Schulman et al., 2015b, GAE) with a fixed $\lambda_{\text{GAE}} \in [0, 1]$:

$$\delta_t = r_t + \gamma\, V_\phi(s_{t+1}) - V_\phi(s_t), \tag{14}$$

$$A_t = \sum_{l=0}^{T-1-t} (\gamma \lambda_{\text{GAE}})^l \, \delta_{t+l}. \tag{15}$$

The critic is then trained by minimizing the mean squared error over the batch:

$$\mathcal{L}_{\text{critic}}(\phi) = \frac{1}{T} \sum_{t=0}^{T-1} \big(R_t - V_\phi(s_t)\big)^2,$$

using the Adam optimizer with a learning rate of $\alpha$, where $R_t = A_t + V_\phi(s_t)$ is the Monte-Carlo return target. The actor is updated by a single natural policy gradient step obtained from the matrix-free Fisher inverse. Since our method is based on Sherman-Morrison formula, we call the resulting algorithm Sherman-Morrison Actor-Critic (SMAC), which is detailed in Algorithm 1.

## 5 Experimental Setup

To evaluate the performance of our proposed approach, SMAC, we provide experiments in several environments and compare against four baselines. We begin with three simple control environments, before moving on to six MuJoCo tasks (Todorov et al., 2012). For all experiments, we used the Gymnasium framework (Towers et al., 2024).

We implement SMAC on top of the A2C algorithm. In all environments, we evaluate SMAC against three baselines. To measure the benefits of using our NPG approximator, we consider an algorithm that is similar to SMAC in all aspects, except that it uses first-order updates instead of natural gradients. This is equivalent to A2C with a simple stochastic gradient descent (SGD) optimizer, which we denote as AC-SGD. To ensure a fair comparison to vanilla A2C, we also compare against the AC-Adam baseline, where A2C is trained

---

**Algorithm 1** SM-ActorCritic

---

1: **input:** steps per update $T$, step size $\eta$, damping $\lambda$
2: initialise $\theta$, $\phi$
3: **for** iteration $k = 0, 1, \ldots$ **do**
4:     rollout $T$ steps, store $(s^k, a^k, r^k)$
5:     compute advantages $A_k$ with GAE
6:     **for all** samples $k$ **do**
7:         $g^k \leftarrow \nabla_\theta \log \pi_\theta(a \,|\, s) A(s, a)$
8:         $\theta^{k+1} \leftarrow \theta^k + \eta \, \hat{\boldsymbol{F}}^{k^{-1}} g^k$ with equation 5
9:     **end for**
10:    update critic: $\phi^{k+1} \leftarrow \phi^k - \alpha \nabla_\phi \sum_t (R_t^k - V_\phi(s_t))^2$
11: **end for**

---

using the more complex Adam optimizer. We also test SMAC against other NPG approximators. The AC-CG baseline uses the conjugate gradient algorithm to directly compute the NPG during the A2C update, without ever computing or storing the Fisher information matrix itself, similarly to TRPO (Schulman et al., 2015a). Finally, the second NPG approximator uses the Kronecker-factored approximate curvature (Martens & Grosse, 2015, KFAC). Our implementation, AC-KFAC, follows the application of KFAC to actor-critic algorithms as proposed by Wu et al. (2017), but omits the trust region.

SMAC and all the baselines we compare against compute advantages using the GAE. This is because GAE handles the bias-variance trade-off better than the simpler advantage estimator used by A2C, and has been shown to be effective in high-dimensional continuous control (Schulman et al., 2015b).

For each of the five algorithms in each environment, we train five models using different random seeds. The results reported throughout training are averaged over these seeds, with standard deviations included. We evaluate agent performance using (undiscounted) returns, averaged across episodes. Besides the maximum average returns achieved, we are also interested in convergence speed.

Furthermore, we also track the average log-probabilities of the actions taken. We treat these as a measure of the agents' confidence in their actions. We therefore expect a well-performing policy to become more deterministic, achieving both higher log-probabilities and higher returns. On the contrary, we can also diagnose overconfident policies, where an increase in log-probabilities does not correlate to an increase in returns. The agent may therefore prematurely converge to an overly deterministic sub-optimal policy, which is likely to hinder exploration and thus future improvements. We report the log-probabilities measured in the supplementary material.

In all environments, episodes have a horizon of 1000 timesteps. To increase the readability of the plots, we group timesteps into 100 bins with no overlap and report the average of each bin. We apply additional smoothing by computing an exponentially weighted mean over the bins, with a smoothing factor of 0.1. The average performances and standard deviations we report are then computed from these binned and smoothed results. However, for the sake of completeness, we also plot the raw, unbinned, and unsmoothed results in the supplementary material.

## 5.1 Practical Method

The matrix-free updates operate on a single trajectory sample. Although the per-step cost is only $\mathcal{O}(d)$, when applying this per-sample scheme to a full batch of size $b$, the update must be executed b times, so the effective time complexity for the batch becomes $\mathcal{O}(bd)$, so the computational cost can be high to update a whole batch.

|  | AC-SGD | AC-Adam | AC-CG | AC-KFAC | SMAC (Ours) |
|---|---|---|---|---|---|
| Acrobot | $-\mathbf{86.2} \pm \mathbf{4.7}$ | $-88.4 \pm 5.7$ | $-93.3 \pm 11.6$ | $-170.7 \pm 16.5$ | $-94.1 \pm 17.0$ |
| Cartpole | $962.3 \pm 38.5$ | $\mathbf{989.5} \pm \mathbf{14.1}$ | $977.9 \pm 34.4$ | $331.4 \pm 142.3$ | $973.4 \pm 49.5$ |
| Pendulum | $-2116 \pm 678$ | $-5731 \pm 272$ | $-\mathbf{190} \pm \mathbf{23}$ | $-1161 \pm 15$ | $-2226 \pm 1138$ |
| Half Cheetah | $-1672 \pm 10.6$ | $-646 \pm 9.7$ | $1766 \pm 864$ | $513 \pm 50.3$ | $\mathbf{2936} \pm \mathbf{946}$ |
| Hopper | $180.5 \pm 9.1$ | $303.8 \pm 23.9$ | $172.7 \pm 10.9$ | $305.0 \pm 33.6$ | $\mathbf{331.2} \pm \mathbf{20.9}$ |
| Swimmer | $12.4 \pm 3.0$ | $17.3 \pm 0.9$ | $23.8 \pm 1.7$ | $24.2 \pm 2.5$ | $\mathbf{28.5} \pm \mathbf{1.4}$ |
| Walker | $213.4 \pm 36.9$ | $79.3 \pm 58.7$ | $\mathbf{229.8} \pm \mathbf{38.3}$ | $83.7 \pm 72.9$ | $204.1 \pm 58.0$ |
| Humanoid | $4539 \pm 171$ | $3105 \pm 345$ | $3175 \pm 1445$ | $260.8 \pm 11.15$ | $\mathbf{4625} \pm \mathbf{274}$ |
| Pusher | $-433.6 \pm 41.6$ | $-568.3 \pm 16.4$ | $-441.0 \pm 35.6$ | $-1086.3 \pm 16.2$ | $-\mathbf{408.2} \pm \mathbf{31.5}$ |

Table 1: Average returns per episode, measured at the end of training, for each task in both classic control and MuJoCo environments. All results are averaged over 5 seeds ($\pm$ standard deviations).

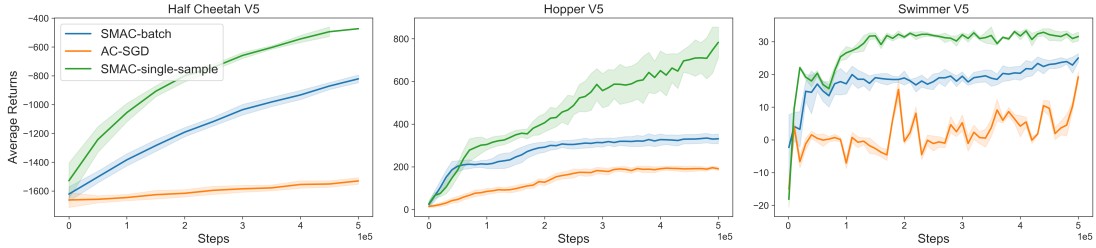

Figure 3: Effect of Batch Size.

To investigate this trade-off, we evaluate an alternative that uses the average gradient of log-probability over all sampled state–action transitions in the batch:

$$\bar{\ell} = \frac{1}{B} \sum_{i=1}^{B} \nabla_\theta \log \pi_\theta(a_i \mid s_i).$$

Where $B$ denotes the total number of state-action pairs collected in the batch. The regularized empirical Fisher then becomes

$$\tilde{\boldsymbol{F}} = \lambda \boldsymbol{I} + \bar{\ell}^k \, (\bar{\ell}^k)^\top,$$

This is a rank-one moment approximation that assumes vectors in batch have small angular variance. The bias can be written explicitly:

$$\mathbb{E}[\tilde{\boldsymbol{F}} - \hat{\boldsymbol{F}}] = \frac{1}{B} Cov(\ell) \succeq 0$$

so $\tilde{\boldsymbol{F}}$ underestimates $\hat{\boldsymbol{F}}$ by a positive semidefinite term that shrinks as gradients concentrate or as $B$ grows.

and the update direction of SMAC is given by

$$\theta^{k+1} = \theta^k - \eta \left[ \frac{1}{\lambda} g^k - \frac{\bar{\ell}^k \, (\bar{\ell}^k)^\top g^k}{\lambda^2 + \lambda (\bar{\ell}^k)^\top \bar{\ell}^k} \right].$$

**Results**  Fig. 3 compares the single-sample and batch-mean variants on Mujoco HalfCheetah and Hopper. Using $B = 1000$ reduces the overall optimization time by roughly 80% while achieving returns that remain close to the single-sample baseline and clearly outperform the standard A2C optimized with SGD. Therefore, to further balance computational resources and performance, we implement the version with a batch size of 1000 in all subsequent experiments.

## 5.2 Effect of damping coefficient $\lambda$

We use the EF $\hat{\boldsymbol{F}}^k = \lambda \boldsymbol{I} + \ell^k(\ell^k)^\top$ with $\lambda > 0$. After applying the Sherman-Morrison formula, its inverse has the closed form

$$(\lambda \boldsymbol{I} + \ell^k(\ell^k)^\top)^{-1} = \frac{1}{\lambda}(\boldsymbol{I} - \frac{\ell^k(\ell^k)^\top}{\lambda + (\ell^k)^\top \ell^k}).$$

It scales directions unevenly, components perpendicular to $\ell$ are reduced by a factor $\frac{1}{\lambda}$, while the component along $\ell$ is reduced by $\frac{1}{\lambda + (\ell^k)^\top \ell^k}$.

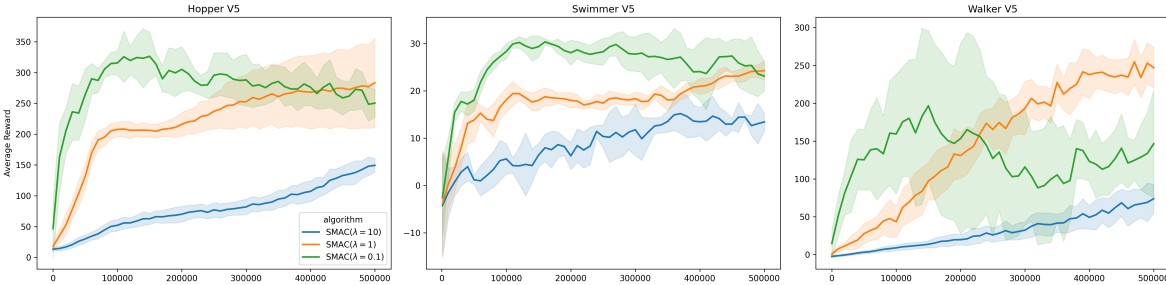

Figure 4: Effect of different $\lambda$

**Results.** We evaluate SMAC on three Mujoco tasks, compare three damping coeffcient value $\lambda \in \{0.1, 1, 10\}$. Overall, the results agree with our analysis: too large $\lambda$ slows learning, too small $\lambda$ can make the update noisy, and a moderate $\lambda$ gives the best stability–speed tradeoff on average. The choices in practice are more complex, for example, the choice of $\lambda$ is coupled with the learning rate because both scale the step, so a balance is needed. A feasible path is to use a small damping coefficient to keep the EF term close to its undamped form, and then apply a normalization so the update magnitude is preserved.

## 5.3 Effect of previous estimate EF

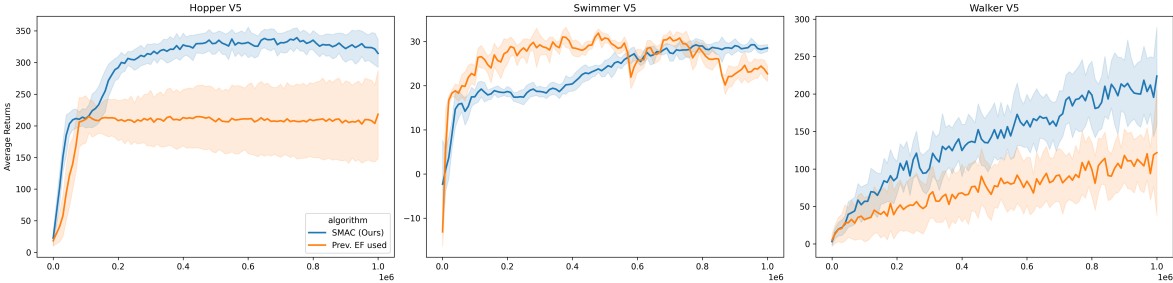

Figure 5: Previous EF used vs. Ours

Using the previous EF not only increases the memory and time complexity to $\mathcal{O}(d^2)$, but introduces a lag bias: after each update the policy and state distribution change, so the old EF reflects the old distribution and can mis-scale directions when curvature changes quickly. In this experiments we used a mixed estimator $\boldsymbol{F}^k = \beta \boldsymbol{F}^{k-1} + (1 - \beta)\ell^k(\ell^k)^\top$ with a fixed $\beta = 0.5$ to limit reliance on the old EF. Results show that on Swimmer it gives faster early improvement but remains less stable, while on Hopper and Walker it learns more slowly with larger variance. Therefore, if the KL change is strictly constrained, using the previous estimate EF is still feasible, and the choice of $\beta$ should depend on the rate of policy change.

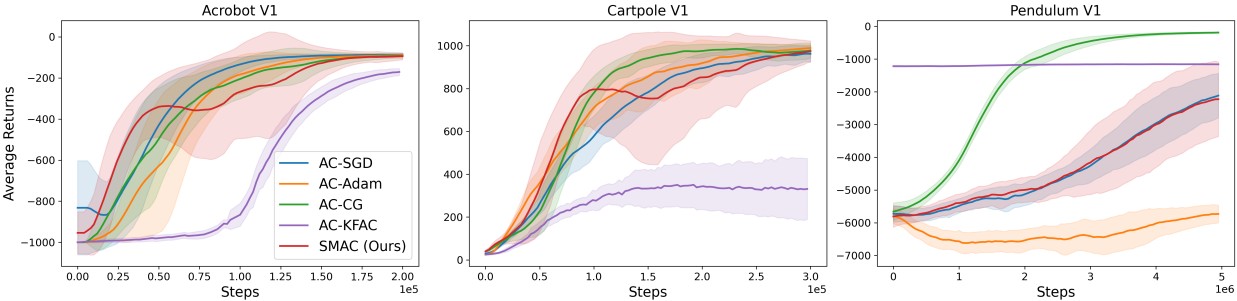

Figure 6: Results on three classic control tasks. We report the average return per episode. All results are first grouped into bins and smoothed, and then
averaged over 5 random seeds, with shaded areas showing standard deviation.

## 6 Results

We provide the final performance achieved by training each of the five algorithms on each of the nine tasks. The results in Table 1 show that our method, SMAC, can achieve competitive results in two of the three simple control tasks. More importantly, the benefits of SMAC appear to increase in more complex environments, such as MuJoCo, where SMAC outperforms the other baselines in terms of final returns in all but one task. A more in-depth analysis of these results, with a special focus on convergence rate, is given in this section.

### 6.1 Classic Control

In the classic control environment, our method, SMAC, reaches a high initial average return faster than the baselines on two of the tasks, as shown in Fig. 6 (see Fig. 8 for raw results). This, however, comes at the cost of additional variance across seeds. The overall performance of SMAC is also competitive in those two tasks.

In *Acrobot*, SMAC only requires 40000 timesteps to pass the average return threshold of $-400$. This is faster than AC-Adam, AC-SGD, AC-CG, and AC-KFAC, which require 70%, 35%, 50%, and 235% more timesteps, respectively. This accelerated learning is, however, followed by unstable training until approximately 125000 timesteps. Despite this, SMAC stabilizes and converges, reaching a final performance of $-94.1 \pm 6.3$, which is similar to three of the baselines, and higher than AC-KFAC.

A similar behavior can be observed in *Cartpole*. SMAC agents achieve 75% of the maximum return possible in 87000 timesteps, slightly faster than AC-CG, which needs approximately 10% more timesteps. AC-Adam and AC-SGD are even slower, requiring 28% and 62% more timesteps, respectively, to reach the same threshold. At the same time, the average performance of AC-KFAC never improves above 40% of the optimal policy's return. SMAC becomes more unstable after this initial period, but stabilizes and reaches a competitive performance of $973.4 \pm 29.8$ by the end of training.

In contrast, in the last environment, *Pendulum*, our method fails to outperform AC-CG in terms of convergence speed and overall performance. AC-CG requires only 1.5 million timesteps to match the final performance of SMAC, indicating that our proposed approximation of the FIM can struggle to improve learning in some environments. Moreover, SMAC is also outperformed by AC-KFAC, which quickly finds a good policy but then stops learning. However, SMAC still achieves a final return of $-2226 \pm 859$, outperforming AC-Adam and matching AC-SGD, with the former also displaying much slower and unstable learning.

As proposed in Sec. 5, the validity of our results is also confirmed by the action log-probabilities recorded (Fig. 10). In *Acrobot* and *Cartpole*, the two tasks in which SMAC performs competitively, the log-probabilities converge faster than those of the baseline methods. Together with the increase in performance achieved by

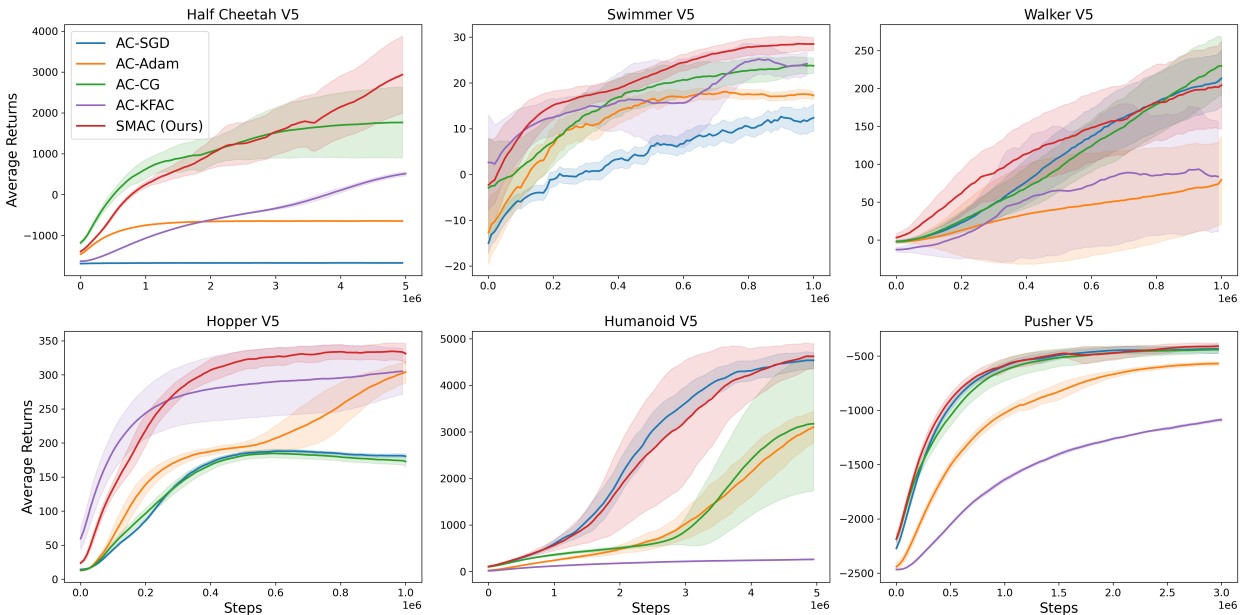

Figure 7: Results on six MuJoCo tasks. We report the average return per episode. All results are first grouped into bins and smoothed, and then averaged over 5 random seeds, with shaded areas showing standard deviation.

SMAC in these tasks, this may indicate that our agents can rapidly become confident in their policies when taking actions that yield increasingly higher returns. At the same time, in *Pendulum*, the log-probabilities of SMAC match those of AC-SGD and are bounded above by AC-CG. This could indicate that our method becomes overconfident in its actions too early, finally leading to a sub-optimal policy.

## 6.2 MuJoCo

In the more complex MuJoCo environment, we have found that throughout training, our method, SMAC, consistently outperforms or matches the baseline algorithms in terms of performance in 5 out of 6 tasks, as shown in Fig. 7 (see Fig. 9 for raw results). In the sixth task, SMAC still achieves comparable results.

SMAC shows improvements over existing methods in *Half Cheetah*, where it achieves an average return of $2936 \pm 946$, surpassing the next-best algorithm, AC-CG, by over 1100 points. While it learns more slowly for the first 2 million timesteps, SMAC ultimately matches the final performance of AC-CG after only 3.4 million timesteps, which represents just 68% of the training budget. It then continues to improve, while AC-CG plateaus. This superior performance comes at the cost of lower stability, with both SMAC and AC-CG being much more unstable than AC-KFAC, and the simpler AC-Adam and AC-SGD.

Similar trends can be observed in *Hopper*, where SMAC achieves the final performance of AC-Adam using almost three times fewer timesteps. Our method achieves a final average return of $331.2 \pm 29.7$, outperforming the second-best performances of AC-Adam and AC-KFAC by approximately 30 points. The stability of SMAC appears to be comparable to that of AC-Adam, higher than that of AC-KFAC, but lower than that of the other two baselines.

Likewise, in *Swimer*, SMAC consistently outperforms the second-best baselines, with a final average return of $28.5 \pm 0.8$, which is approximately 5 points higher than those of AC-CG and AC-KFAC. Moreover, it learns the fastest, matching the peak performance of AC-CG almost twice as fast. The stability of SMAC is also greater than that of AC-SGD, AC-CG, and AC-KFAC, and competitive with that of AC-Adam.

In *Walker*, our method learns faster than the baselines in the first 60% of the timesteps. While its final performance of $204.1 \pm 87.9$ is lower than those of AC-CG and AC-SGD, which achieve $229.8 \pm 23.9$ and $213.4 \pm 42.9$, respectively, SMAC reaches the threshold of 115 (half of the final return of AC-CG) in only 410000 timesteps, while AC-CG itself needs 39% more. Additionally, our approach outperforms AC-Adam and AC-KFAC in both average return and stability.

For the *Humanoid* environment, SMAC achieves the top final performance of $4625 \pm 277$ average return, followed closely by AC-SGD with $4539 \pm 110$. It greatly outperforms AC-Adam and AC-CG in terms of both final returns and final stability, with only AC-SGD and AC-KFAC being more stable. However, the performance of AC-KFAC is much lower than that of any of the other algorithms. While AC-SGD is slightly faster than our method, SMAC still learns much faster than AC-Adam and AC-CG, matching their final performance after using just 59% of the total allocated training budget.

In *Pusher*, the performance of SMAC closely follows that of AC-SGD, slightly outperforming the latter with a final average return of $-408.2 \pm 32.2$. Our method is also more sample efficient, passing the $-441$ return threshold (final performance of AC-CG) 21% faster than AC-CG and 9% faster than AC-SGD. While all agents stabilize by the end of training, only AC-Adam and AC-KFAC end up being more stable than SMAC, with AC-SGD having the highest standard deviation.

Finally, we turn our attention to the average log-probabilities of the actions taken during training (Fig. 11), as proposed in Sec. 5. The increase in log-probabilities shown by SMAC is stable and correlates with the increase in average returns given in Fig. 7. This may indicate that the policy quickly becomes confident in the actions it chooses, without becoming overconfident. Note that the log-probabilities of our method converge the fastest in 2 out of 6 tasks. While the log-probabilities of AC-CG converge first in *Humanoid* and *Pusher*, this does not lead to better performance. On the contrary, in the former task, AC-CG agents become overconfident in their sub-optimal actions. While in *Half Cheetah* the log-probabilities of AC-CG converge first, SMAC ultimately achieves a higher confidence in its actions, which matches its superior final performance. A similar trend appears in *Walker*, with AC-KFAC converging the fastest, but to a sub-optimal policy. The slow convergence of the other two agents' log-probabilities in most tasks may indicate uncertainty in their actions, even when their performance matches that of SMAC.

## 7    Conclusion

In this work, we introduced a simple yet effective rank-1 approximation to the Natural Policy Gradient in the Actor-Critic framework. By using a regularized Empirical Fisher matrix and the Sherman–Morrison formula, we enable scalable and efficient natural policy gradient updates with only $\mathcal{O}(d)$ complexity. We theoretically prove the convergence properties of this approximation and integrate it into the standard Actor-Critic algorithm to demonstrate its practical utility. Experimental results on classic control and MuJoCo tasks show that our method achieves both fast and stable convergence in most environments, outperforms methods such as Advantage Actor-Critic and conjugate gradient in TRPO in several environments, offering a promising alternative for efficient NPG approximations.

### Broader Impact

Reinforcement learning is now used in many domains, from fine-tuning language models to controlling systems such as power grids or data centres, and this broad use can bring both benefits and risks. It can improve efficiency, safety, and energy use, but it can also amplify existing biases or support decision systems that have harmful social or economic effects. Our work focuses only on the optimisation aspect: we study how to solve a given policy optimisation problem faster, so that a fixed objective is reached in fewer gradient steps. If the training data for language models or the environment model for control tasks reflects real-world dynamics and values accurately, a more efficient optimiser can produce better policies with less computation and a smaller training carbon footprint. At the same time, any negative societal impact would mainly arise from biased, incomplete, or misaligned data, objectives, and reward designs, rather than from the particular optimization method used.

**Acknowledgment**

SPD was funded by the Dean's Doctoral Scholarship at the University of Manchester. SK acknowledges funding from the UKRI Turing AI World-Leading Researcher Fellowship (EP/W002973/1). MS acknowledges funding from AI Hub in Generative Models by Engineering & Physical Sciences Research Council (EPSRC) with Funding Body Ref: EP/Y028805/1.

**Code availability**

Our code is publicly available at `https://github.com/agent-lab/sherman-morrison-actor-critic`.

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

# A Appendix

## A.1 Notations.

Let $\gamma \in (0,1)$ be the discount, $r : \mathcal{S} \times \mathcal{A} \to [-R, R]$ is the reward function, and $H \in \mathbb{N} \cup \{\infty\}$ the horizon. Define the truncated objective

$$J^H(\theta) = \mathbb{E}_{\tau \sim \pi_\theta} \Big[ \sum_{t=0}^{H-1} \gamma^t r(s_t, a_t) \Big], \qquad J(\theta) = J^\infty(\theta). \tag{16}$$

For each trajectory $\tau^H = (s_0, a_0, \ldots, s_{H-1}, a_{H-1})$ sampled under $\pi_\theta$, let $g(\tau^H \mid \theta)$ denote a single-trajectory unbiased estimator of the truncated policy gradient:

$$\mathbb{E}\big[g(\tau^H \mid \theta)\big] = \nabla J^H(\theta). \tag{17}$$

$$g(\tau^H \mid \theta) = \sum_{t=0}^{H-1} \gamma^t G_t \nabla_\theta \log \pi_\theta(a_t \mid s_t) \tag{18}$$

## A.2 Assumptions

**Assumption 1.** *Let $g(\tau^H|\theta)$ be an unbiased single-trajectory estimator of $\nabla J^H(\theta)$ Assume there exists $\sigma > 0$ such that for all $\theta$,*

$$\mathbb{E}\Big[\big\| g(\tau^H \mid \theta) - \mathbb{E}\big[g(\tau^H \mid \theta)\big] \big\|^2\Big] \leq \sigma^2,$$

**Assumption 2.** *The Fisher Information matrix induced by the policy $\pi_\theta$ is Positive Definite and satisfies*

$$\boldsymbol{F}(\theta) = \mathbb{E}_{s \sim d_\rho^{\pi_\theta}} \mathbb{E}_{a \sim \pi_\theta} \left[ \nabla_\theta \log \pi_\theta \cdot \nabla_\theta \log \pi_\theta^T \right] \succcurlyeq \mu_F \mathbf{I} \tag{19}$$

**Assumption 3.** *The policy gradient estimates are bounded by a constant $G \geq 0$ and the change in policy gradient at different timestep are bounded by the change in policy parametrization weights. So,*

$$\|\nabla_\theta \log \pi_\theta\| \leq G \tag{20}$$

$$\|\nabla_\theta \log \pi_{\theta_1} - \nabla_\theta \log \pi_{\theta_2}\| \leq M \|\theta_1 - \theta_2\| \tag{21}$$

**Assumption 4.** ***Transferrable Compatible Function Approximator****: When we approximate the Advantage function using the policy $\pi_\theta$ it induces a transfer error defined as $\epsilon_{bias}$ which is zero for softmax parametrization and is very small for dense neural policy class.*

$$L_{\nu^\star}(w_\star^\theta; \theta) = \mathbb{E}_{(s,a) \sim \nu^\star} \left[ \left( A^{\pi_\theta}(s,a) - (1-\gamma)(\boldsymbol{w}_\star^\theta)^\top \nabla_\theta \log \pi_\theta(a|s) \right)^2 \right] \leq \varepsilon_{bias} \tag{22}$$

*where $\nu^\star(s,a) = d_\rho^{\pi^\star}(s) \cdot \pi^\star(s,a)$ is the state-action distribution induced by an optimal policy $\pi^\star$ and $\boldsymbol{w}_\star^\theta = \operatorname{argmin}_w L_{\nu_\rho^{\pi_\theta}}(\boldsymbol{w}; \theta)$ is obtained via the full natural policy gradient direction at the parametrization $\theta$.*

**Remark 1.** *Regarding $\boldsymbol{F}(\theta)$, we know from Assumptions 1 and 2 that*

$$\mu_F I_d \preccurlyeq \boldsymbol{F}(\theta) \preccurlyeq G^2 I_d \text{ for any } \theta \in \mathbf{R}^d.$$

**Remark 2.** *Now we can get an upper bound to the second term of Equation 4 using Remark 1 as:*

$$\left\| \frac{\hat{\boldsymbol{F}}^k}{\lambda^2 + \lambda Tr(\hat{\boldsymbol{F}}^k)} \right\| \leq \frac{G^2}{\lambda^2 + \lambda \mu_F}.$$

**Remark 3.** *Assumption 1 treats $\sigma^2$ as a constant upper bound on the variance of the truncated policy gradient estimator, although in general $\mathrm{Var}(g(\tau^H \mid \theta))$ may depend on the horizon $H$. More precisely, for each finite $H$ we write*

$$\mathrm{Var}\big(g(\tau^H \mid \theta)\big) = \sigma_H^2.$$

*For likelihood–ratio estimators such as REINFORCE, existing analyses show that $\sigma_H^2$ can grow with $H$, but remains finite under smooth Gaussian policies with bounded rewards and features (Zhao et al., 2012; Pirotta et al., 2013; Baxter & Bartlett, 2001). In our proof we use a finite truncated horizon and define*

$$\sigma^2 := \sup_{1 \leq H \leq H_{\max}} \sigma_H^2 < \infty.$$

*This form of bounded–variance assumption is standard in recent convergence analyses of policy gradient methods (Liu et al., 2022; Xu et al., 2020; 2019).*

### A.3 Helper Lemmas

We will establish some helper lemma to prove the global convergence of Sherman-Morrison policy update.

**Lemma 1.** *(Liu et al., 2022) In the stochastic PG update, we have*

$$\frac{1}{K} \sum_{k=0}^{K-1} \mathbb{E}[\|\nabla J^H(\theta^k)\|^2] \leq \frac{\frac{J^{H\star} - J^H(\theta_0)}{K} + \left(\frac{\eta}{2} + L_J \eta^2\right)\frac{\sigma^2}{N}}{\frac{\eta}{2} - L_J \eta^2} \tag{23}$$

*In total, stochastic PG samples $\mathcal{O}\left(\frac{\sigma^2}{(1-\gamma)^2 \varepsilon^2}\right)$ trajectories.*

**Lemma 2.** *(Liu et al., 2022) Let $J(\theta), J^H(\theta)$ are $L_J$-smooth, where $L_J = \frac{MR}{(1-\gamma)^2} + \frac{2G^2 R}{(1-\gamma)^3}$ and*

$$\|\nabla J^H(\theta) - \nabla J(\theta)\| \leq GR \left(\frac{H+1}{1-\gamma} + \frac{\gamma}{(1-\gamma)^2}\right)\gamma^H$$

**Lemma 3.** *(Liu et al., 2022) Let $\boldsymbol{w}_\star^k = F^{-1}(\theta^k)\nabla J(\theta^k)$ be the exact NPG update direction at $\theta^k$. Then, we have*

$$J(\pi^\star) - \frac{1}{K}\sum_{k=0}^{K-1} J(\theta^k) \leq \frac{\sqrt{\varepsilon_{bias}}}{1-\gamma} + \frac{1}{\eta K}\mathbb{E}_{s \sim d_\rho^{\pi^\star}}\left[KL\left(\pi^\star(\cdot|s)||\pi_{\theta^0}(\cdot|s)\right)\right]$$
$$+ \frac{G}{K}\sum_{k=0}^{K-1}\|\boldsymbol{w}^k - \boldsymbol{w}_\star^k\| + \frac{M\eta}{2K}\sum_{k=0}^{K-1}\|\boldsymbol{w}^k\|^2. \tag{24}$$

### A.4 Global Convergence of SM Update

Let us take $\boldsymbol{w}^k$ as the update direction of Sherman-Morrison. To this end, we need to upper bound $\frac{1}{K}\sum_{k=0}^{K-1}\|\boldsymbol{w}^k - \boldsymbol{w}_\star^k\|$, $\frac{1}{K}\sum_{k=0}^{K-1}\|\boldsymbol{w}^k\|^2$, and $\frac{1}{K}\mathbb{E}_{s \sim d_\rho^{\pi^\star}}\left[KL\left(\pi^\star(\cdot|s)||\pi_{\theta^0}(\cdot|s)\right)\right]$, where $\boldsymbol{w}_\star^k = F^{-1}(\theta^k)\nabla J(\theta^k)$ is the exact NPG update direction at $\theta^k$.

- Bounding $\frac{1}{K}\sum_{k=0}^{K-1}\|\boldsymbol{w}^k - \boldsymbol{w}_\star^k\|$.

We know from Jensen's inequality that $\left(\mathbb{E}[\|\boldsymbol{w}_t^{j+1} - \boldsymbol{w}_{t,\star}^{j+1}\|]\right)^2 \leq \mathbb{E}[\|\boldsymbol{w}_t^{j+1} - \boldsymbol{w}_{t,\star}^{j+1}\|^2]$ so,

$$
\begin{aligned}
&\left(\frac{1}{K}\sum_{k=0}^{K-1}\mathbb{E}[\|\boldsymbol{w}^k - \boldsymbol{w}_\star^k\|]\right)^2 \\
&\leq \frac{1}{K}\sum_{k=0}^{K-1}\left(\mathbb{E}[\|\boldsymbol{w}^k - \boldsymbol{w}_\star^k\|]\right)^2 \\
&\leq \frac{1}{K}\sum_{k=0}^{K-1}\mathbb{E}[\|\boldsymbol{w}^k - \boldsymbol{w}_\star^k\|^2] \\
&\leq \frac{2}{K}\sum_{k=0}^{K-1}\mathbb{E}[\|\boldsymbol{w}^k - \nabla J(\theta^k)\|^2] + \frac{2}{K}\sum_{k=0}^{K-1}\mathbb{E}[\|\nabla J(\theta^k) - \boldsymbol{w}_\star^k\|^2]
\end{aligned}
\tag{25}
$$

Let

$$
\boldsymbol{g}^k = \frac{1}{N}\sum_{i=1}^{N} g(\tau_i^H|\theta^k),
$$

be the PG direction and

$$
\boldsymbol{w}^k = \frac{1}{\lambda}\boldsymbol{g}^k - \frac{\hat{\boldsymbol{F}}^k}{\lambda^2 + \lambda Tr(\hat{\boldsymbol{F}}^k)}\boldsymbol{g}^k
$$

be the weight update from the Sherman-Morrison update then we have from Lemma 2 and Remark 2 and Assumption 3 that

$$
\begin{aligned}
&\frac{1}{K}\sum_{k=0}^{K-1}\mathbb{E}[\|\boldsymbol{w}^k - \nabla J(\theta^k)\|^2] \\
&= \frac{1}{K}\sum_{k=0}^{K-1}\mathbb{E}\left[\left\|\frac{1}{\lambda}\boldsymbol{g}^k - \frac{\hat{\boldsymbol{F}}^k}{\lambda^2 + \lambda Tr(\hat{\boldsymbol{F}}^k)}\boldsymbol{g}^k - \nabla J(\theta^k)\right\|^2\right] \\
&= \frac{1}{K}\sum_{k=0}^{K-1}\mathbb{E}\left[\left\|\frac{1}{\lambda}(\boldsymbol{g}^k - \nabla J^H(\theta^k)) - \frac{\hat{\boldsymbol{F}}^k}{\lambda^2 + \lambda Tr(\hat{\boldsymbol{F}}^k)}(\boldsymbol{g}^k - \nabla J^H(\theta^k)) - (\nabla J(\theta^k) - \nabla J^H(\theta^k))\right.\right. \\
&\qquad\left.\left. + \frac{1}{\lambda}\nabla J^H(\theta^k) - \frac{\hat{\boldsymbol{F}}^k}{\lambda^2 + \lambda Tr(\hat{\boldsymbol{F}}^k)}\nabla J^H(\theta^k) - \nabla J^H(\theta^k)\right\|^2\right] \\
&\leq 4\frac{\sigma^2}{N}\left(\frac{1}{\lambda^2} + \frac{G^2}{\lambda^2 + \lambda\mu_F}\right) + 4G^2R^2\left(\frac{H+1}{1-\gamma} + \frac{\gamma}{(1-\gamma)^2}\right)^2\gamma^{2H} \\
&\quad + 4\left(1 + \frac{1}{\lambda^2} + \frac{G^2}{\lambda^2 + \lambda\mu_F}\right)\frac{1}{K}\sum_{k=0}^{K-1}\mathbb{E}[\|\nabla J^H(\theta^k)\|^2]
\end{aligned}
\tag{26}
$$

Furthermore, Assumption 2 tells us that

$$
\begin{aligned}
&\mathbb{E}[\|\nabla J(\theta^k) - \boldsymbol{w}_\star^k\|^2] \\
&= \mathbb{E}[\|\nabla J(\theta^k) - F^{-1}(\theta^k)\nabla J(\theta^k)\|^2] \\
&\leq \left(1 + \frac{1}{\mu_F}\right)^2\mathbb{E}[\|\nabla J(\theta^k)\|^2] \\
&\leq \left(1 + \frac{1}{\mu_F}\right)^2\left(2\mathbb{E}[\|\nabla J^H(\theta^k)\|^2] + 2G^2R^2\left(\frac{H+1}{1-\gamma} + \frac{\gamma}{(1-\gamma)^2}\right)^2\gamma^{2H}\right).
\end{aligned}
\tag{27}
$$

Combining equation 26 and equation 27 with equation 25 gives

$$
\begin{aligned}
&\frac{1}{K} \sum_{k=0}^{K-1} \mathbb{E}[\|\boldsymbol{w}^k - \boldsymbol{w}_\star^k\|] \\
&\leq \left( 4\frac{\sigma^2}{N} \left( \frac{1}{\lambda^2} + \frac{G^2}{\lambda^2 + \lambda\mu_F} \right) + 4G^2 R^2 \left( \frac{H+1}{1-\gamma} + \frac{\gamma}{(1-\gamma)^2} \right)^2 \gamma^{2H} \left( 1 + \left( 1 + \frac{1}{\mu_F} \right)^2 \right) \right. \\
&\left. \quad + 4 \left( 1 + \left( 1 + \frac{1}{\mu_F} \right)^2 + \frac{1}{\lambda^2} + \frac{G^2}{\lambda^2 + \lambda\mu_F} \right) \frac{1}{K} \sum_{k=0}^{K-1} \mathbb{E}[\|\nabla J^H(\theta^k)\|^2] \right)^{0.5}
\end{aligned}
\tag{28}
$$

And recall from equation 23 that

$$
\frac{1}{K} \sum_{k=0}^{K-1} \mathbb{E}[\|\nabla J^H(\theta^k)\|^2] \leq \frac{\frac{J^{H,\star} - J^H(\theta_0)}{K} + (\frac{\eta}{2} + L_J \eta^2)\frac{\sigma^2}{N}}{\frac{\eta}{2} - L_J \eta^2}.
$$

Let us take $\eta = \frac{1}{4L_J}$. Then we get

$$
\frac{1}{K} \sum_{k=0}^{K-1} \mathbb{E}[\|\nabla J^H(\theta^k)\|^2] \leq \frac{16 L_J (J^{H,\star} - J^H(\theta_0))}{K} + \frac{3\sigma^2}{N}
$$

In addition, let $H$, $N$, and $K$ satisfy

$$
\begin{aligned}
&\frac{1}{3}(\frac{\varepsilon}{3G})^2 \geq 4G^2 R^2 \left( \frac{H+1}{1-\gamma} + \frac{\gamma}{(1-\gamma)^2} \right)^2 \gamma^{2H} \left( 1 + \left( 1 + \frac{1}{\mu_F} \right)^2 \right)^2 \\
&N \geq \frac{4\sigma^2 \left( 3 + \frac{4}{\lambda^2} + \frac{4G^2}{\lambda^2 + \lambda\mu_F} + 3\left( 1 + \frac{1}{\mu_F} \right)^2 \right)}{\frac{1}{3}\left( \frac{\varepsilon}{3G} \right)^2}, \\
&K \geq \frac{64 L_J (J^{H,\star} - J^H(\theta_0)) \left( \left( 1 + \frac{1}{\mu_F} \right)^2 + 1 + \frac{1}{\lambda^2} + \frac{G^2}{\lambda^2 + \lambda\mu_F} \right)}{\frac{1}{3}\left( \frac{\varepsilon}{3G} \right)^2}.
\end{aligned}
\tag{29}
$$

Then, we have

$$
\frac{G}{K} \sum_{k=0}^{K-1} \mathbb{E}[\|\boldsymbol{w}^k - \boldsymbol{w}_\star^k\|] \leq \frac{\varepsilon}{3}.
\tag{30}
$$

- Bounding $\frac{1}{K} \sum_{k=0}^{K-1} \|\boldsymbol{w}^k\|^2$.

We have from equation 26 and equation 23 that

$$
\frac{1}{K} \sum_{k=0}^{K-1} \mathbb{E}\|\boldsymbol{w}^k\|^2
$$

$$
= \frac{1}{K} \sum_{k=0}^{K-1} \mathbb{E}\left[\left\|\frac{1}{\lambda}\boldsymbol{g}^k - \frac{\hat{\boldsymbol{F}}^k}{\lambda^2 + \lambda Tr(\hat{\boldsymbol{F}}^k)}\boldsymbol{g}^k\right\|^2\right]
$$

$$
= \frac{1}{K} \sum_{k=0}^{K-1} \mathbb{E}\left[\left\|\frac{1}{\lambda}(\boldsymbol{g}^k - \nabla J^H(\theta^k)) - \frac{\hat{\boldsymbol{F}}^k}{\lambda^2 + \lambda Tr(\hat{\boldsymbol{F}}^k)}(\boldsymbol{g}^k - \nabla J^H(\theta^k)) + \frac{1}{\lambda}\nabla J^H(\theta^k)\right.\right.
$$

$$
\left.\left. - \frac{\hat{\boldsymbol{F}}^k}{\lambda^2 + \lambda Tr(\hat{\boldsymbol{F}}^k)}\nabla J^H(\theta^k)\right\|^2\right]
$$

$$
\leq 2\frac{\sigma^2}{N}\left(\frac{1}{\lambda^2} + \frac{G^2}{\lambda^2 + \lambda\mu_F}\right) + 2\left(\frac{1}{\lambda^2} + \frac{G^2}{\lambda^2 + \lambda\mu_F}\right)\frac{1}{K}\sum_{k=0}^{K-1}\mathbb{E}[\|\nabla J^H(\theta^k)\|^2]
$$

$$
\leq 16\frac{\sigma^2}{N}\left(\frac{1}{\lambda^2} + \frac{G^2}{\lambda^2 + \lambda\mu_F}\right) + \frac{32L_J J^{H\star} - J^H(\theta_0)}{K}\left(\frac{1}{\lambda^2} + \frac{G^2}{\lambda^2 + \lambda\mu_F}\right).
$$

Where to get to the last inequality we have taken $\eta = \frac{1}{4L_J}$ and applied Lemma 1.

$$
N \geq \frac{16\eta\sigma^2\left(\frac{1}{\lambda^2} + \frac{G^2}{\lambda^2 + \lambda\mu_F}\right)}{\frac{\varepsilon}{6}},
$$

$$
K \geq \frac{32L_J\eta(J^{H,\star} - J^H(\theta_0))\left(\frac{1}{\lambda^2} + \frac{G^2}{\lambda^2 + \lambda\mu_F}\right)}{\frac{\varepsilon}{6}},
\tag{31}
$$

we arrive at

$$
\frac{\eta}{K}\sum_{k=0}^{K-1}\mathbb{E}[\|\boldsymbol{w}^k\|^2] \leq \frac{\varepsilon}{3}.
\tag{32}
$$

- Bounding $\frac{1}{K}\mathbb{E}_{s\sim d_\rho^{\pi^\star}}\left[\mathrm{KL}\left(\pi^\star(\cdot|s)||\pi_{\theta^0}(\cdot|s)\right)\right]$.

  By taking

$$
K \geq \frac{3\mathbb{E}_{s\sim d_\rho^{\pi^\star}}\left[\mathrm{KL}\left(\pi^\star(\cdot|s)||\pi_{\theta^0}(\cdot|s)\right)\right]}{\eta\varepsilon}
\tag{33}
$$

we have

$$
\frac{1}{\eta K}\mathbb{E}_{s\sim d_\rho^{\pi^\star}}\left[\mathrm{KL}\left(\pi^\star(\cdot|s)||\pi_{\theta^0}(\cdot|s)\right)\right] \leq \frac{\varepsilon}{3},
\tag{34}
$$

From equation 29

$$
\left(\frac{H+1}{1-\gamma} + \frac{\gamma}{(1-\gamma)^2}\right)^2 \gamma^{2H} \leq C_0\,\varepsilon^2,
\tag{35}
$$

for some constant $C_0 > 0$ that does not depend on $H$ or $\varepsilon$.

For all $H \geq 0$

$$\frac{H+1}{1-\gamma} + \frac{\gamma}{(1-\gamma)^2} \ \le\ \frac{H+1}{1-\gamma} + \frac{1}{(1-\gamma)^2} \ \le\ \frac{C_1(H+1)}{(1-\gamma)^2},$$

for a constant $C_1 \ge 1$ that depends only on $\gamma$. Substituting this into Equation 35 gives

$$(H+1)^2\gamma^{2H} \ \le\ C_2(1-\gamma)^4\,\varepsilon^2, \tag{36}$$

for constant $C_2 > 0$ independent of $H$ and $\varepsilon$.

Since the prefactor grows polynomially in $H$ while $\gamma^{2H}$ decays geometrically,

$$\gamma^{2H} \le c_1\,\varepsilon^2,$$

for some constant $c_1 > 0$ independent of $H$. Taking logarithm on both side yields

$$H = \mathcal{O}\big(\log\big((1-\gamma)^{-1}\varepsilon^{-1}\big)\big). \tag{37}$$

The lower bounds on $N$ in equation 29 and equation 31 have the form

$$N \ \ge\ c_2\sigma^2\left(1 + (1+\tfrac{1}{\mu_F})^2 + \tfrac{1}{\lambda^2} + \tfrac{G^2}{\lambda^2+\lambda\mu_F}\right)\varepsilon^{-2},$$

for a constant $c_2$ that absorbs numerical factors. Since the expression in parentheses is independent of $\varepsilon$, we have

$$N = \mathcal{O}\left(\frac{\sigma^2}{\varepsilon^2}\right). \tag{38}$$

The lower bounds on $K$ collected from equation 29, equation 31, and equation 33 can be written as

$$K \ \ge\ c_3\left(L_J(J^{H,\star} - J^H(\theta_0)) + \mathbb{E}_{s\sim d_\rho^{\pi^\star}}\left[\mathrm{KL}\left(\pi^\star(\cdot|s)\|\pi_{\theta^0}(\cdot|s)\right)\right]\right)\varepsilon^{-2},$$

for a constant $c_3$ depending only on $(1-\gamma)^{-1}$ and the model parameters. Since both terms in parentheses scale as $\mathcal{O}((1-\gamma)^{-2})$. We have

$$K = \mathcal{O}\left(\frac{1}{(1-\gamma)^2\varepsilon^2}\right). \tag{39}$$

In summary, we require $N$, $K$ and $H$ to satisfy equation 29, equation 31, and equation 33, which leads to

$$N = \mathcal{O}\left(\frac{\sigma^2}{\varepsilon^2}\right), \qquad K = \mathcal{O}\left(\frac{1}{(1-\gamma)^2\varepsilon^2}\right), \qquad H = \mathcal{O}\left(\log((1-\gamma)^{-1}\varepsilon^{-1})\right).$$

By combining equation 30, equation 32, equation 34 and equation 24, we can conclude that

$$J(\pi^\star) - \frac{1}{K}\sum_{k=0}^{K-1} J(\theta^k) \le \frac{\sqrt{\varepsilon_{\mathrm{bias}}}}{1-\gamma} + \varepsilon.$$

In total, stochastic Sherman-Morrison Policy gradient requires to sample $KN = \mathcal{O}\left(\frac{\sigma^2}{(1-\gamma)^2\varepsilon^4}\right)$ trajectories.

### A.5 Detailed Derivation of the Toy Problem

The policy is parameterized by a vector $\boldsymbol{\theta} = [\theta_0, \theta_1]^\top$, with independent Bernoulli distribution:

$$\pi_{\boldsymbol{\theta}}(a = 0 \mid x = 0) = \frac{1}{1 + e^{\lambda_p \theta_0}}, \tag{40}$$

$$\pi_{\boldsymbol{\theta}}(a = 0 \mid x = 1) = \frac{1}{1 + e^{\theta_1}}, \tag{41}$$

where $\lambda_p$ scales the first policy parameter and is used to test scaling invariance. If we assume that the agent starts at the state $x = 0$ then the objective (expected return) in this simplified setting is

$$J(\boldsymbol{\theta}) = \pi_{\boldsymbol{\theta}}(a = 0 \mid x = 0) + 2(1 - \pi_{\boldsymbol{\theta}}(a = 0 \mid x = 0))\,\pi_{\boldsymbol{\theta}}(a = 0 \mid x = 1). \tag{42}$$

The policy gradient theorem states:

$$\nabla_{\boldsymbol{\theta}} J(\boldsymbol{\theta}) = \mathbb{E}_{x,a}[\nabla_{\boldsymbol{\theta}} \log \pi_{\boldsymbol{\theta}}(a|x)\, Q(x, a)]. \tag{43}$$

For the Bernoulli parameterization and noting that the policy function resembles sigmoid activation function the derivative of the log–policy can be simplified as

$$\nabla_{\boldsymbol{\theta}} \log \pi_{\boldsymbol{\theta}}(a|x) = \begin{cases} -\lambda_p(1 - \pi_0) & \text{if } x = 0, a = 0 \\ \lambda_p \pi_0 & \text{if } x = 0, a = 1 \\ -(1 - \pi_1) & \text{if } x = 1, a = 0 \\ \pi_1 & \text{if } x = 1, a = 1 \end{cases} \tag{44}$$

where $\pi_0 = \pi_{\boldsymbol{\theta}}(a = 0 \mid x = 0)$ and $\pi_1 = \pi_{\boldsymbol{\theta}}(a = 0 \mid x = 1)$. Using the rewards in the environment, one can derive the analytic policy gradient vector

$$\nabla_{\boldsymbol{\theta}} J(\boldsymbol{\theta}) = \begin{bmatrix} -\lambda_p\, \pi_0(1 - \pi_0)(1 - 2\pi_1) \\ -2(1 - \pi_0)\pi_1(1 - \pi_1) \end{bmatrix}. \tag{45}$$

The geometry of the policy manifold is captured by the FIM and as the states are independent of each other we can write the per state FIM as,

$$F_{ii} = \mathbb{E}_{a \sim \pi(\cdot|x=i)}\left[\nabla_{\theta_i} \log \pi_\theta(a|x = i)\, \nabla_{\theta_i} \log \pi_\theta(a|x = i)^\top\right] \tag{46}$$

So For State (x = 0): The Score functions for $a \in \{0, 1\}$ can be expressed as:

$$\frac{\partial}{\partial \theta_0} \log \pi_\theta(a = 0 \mid x = 0) = -\lambda_p(1 - \pi_0), \qquad \frac{\partial}{\partial \theta_0} \log \pi_\theta(a = 1 \mid x = 0) = \lambda_p \pi_0 \tag{47}$$

Hence,

$$F_{00} = \pi_0[-\lambda_p(1-\pi_0)]^2 + (1-\pi_0)[\lambda_p\pi_0]^2 \tag{48}$$
$$= \lambda_p^2[\pi_0(1-\pi_0)^2 + (1-\pi_0)\pi_0^2] \tag{49}$$
$$= \lambda_p^2\pi_0(1-\pi_0)[(1-\pi_0) + \pi_0] \tag{50}$$
$$= \lambda_p^2\pi_0(1-\pi_0). \tag{51}$$

Similarly for state (x = 1): The Score functions for $a \in \{0,1\}$ ca be expressed as:

$$\frac{\partial}{\partial\theta_1}\log\pi_\theta(a = 0 \mid x = 1) = -(1-\pi_1), \qquad \frac{\partial}{\partial\theta_1}\log\pi_\theta(a = 1 \mid x = 1) = \pi_1 \tag{52}$$

Hence,

$$F_{11} = \pi_1[-(1-\pi_1)]^2 + (1-\pi_1)[\pi_1]^2 \tag{53}$$
$$= \pi_1(1-\pi_1)^2 + (1-\pi_1)\pi_1^2 \tag{54}$$
$$= \pi_1(1-\pi_1)[(1-\pi_1) + \pi_1] \tag{55}$$
$$= \pi_1(1-\pi_1) \tag{56}$$

Owing to independence between states, $F(\boldsymbol{\theta})$ is diagonal:

$$F(\boldsymbol{\theta}) = \begin{bmatrix} \lambda_p^2\pi_0(1-\pi_0) & 0 \\ 0 & \pi_1(1-\pi_1) \end{bmatrix}. \tag{57}$$

Then the NPG update direction is obtained by preconditioning the gradient with the inverse Fisher matrix,

$$\dot{\boldsymbol{\theta}}_{\mathrm{NPG}} = F(\boldsymbol{\theta})^{-1}\nabla_{\boldsymbol{\theta}}J(\boldsymbol{\theta}) = \begin{bmatrix} \dfrac{-(1-2\pi_1)}{\lambda_p} \\ -2(1-\pi_0) \end{bmatrix}. \tag{58}$$

To reduce computation while preserving approximate invariance, the SM approach uses a low–rank inverse metric $\tilde{F}^{-1}$ defined in terms of damping factor $\lambda$ and exact FIM $F(\theta)$ capturing dominant curvature direction,

$$\tilde{F}^{-1} = \frac{1}{\lambda}\left(\boldsymbol{I} - \frac{F(\boldsymbol{\theta})}{\lambda + Tr(F(\boldsymbol{\theta}))}\right) \tag{59}$$

$$\tag{60}$$

Hence the resulting SM update direction can be written as:

$$\dot{\boldsymbol{\theta}}_{\mathrm{SM}} = \tilde{F}^{-1} \nabla_{\boldsymbol{\theta}} J(\boldsymbol{\theta}) \tag{61}$$

$$= \frac{1}{\lambda} \left( \boldsymbol{I} - \frac{F(\boldsymbol{\theta})}{\lambda + Tr(F(\boldsymbol{\theta}))} \right) \nabla_{\boldsymbol{\theta}} J(\boldsymbol{\theta}) \tag{62}$$

$$= \frac{1}{\lambda} \left( \nabla_{\boldsymbol{\theta}} J(\boldsymbol{\theta}) - \frac{F(\boldsymbol{\theta}) \nabla_{\boldsymbol{\theta}} J(\boldsymbol{\theta})}{\lambda + Tr(F(\boldsymbol{\theta}))} \right) \tag{63}$$

$$= \begin{bmatrix} \dfrac{1}{\lambda}(-\lambda_p \, \pi_0(1-\pi_0)(1-2\pi_1)) - \dfrac{\lambda_p^3 \, \pi_0^2(1-\pi_0)^2(1-2\pi_1)}{\lambda^2 + \lambda \left[ \lambda_p^2 \, \pi_0(1-\pi_0) + \pi_1(1-\pi_1) \right]} \\[2ex] \dfrac{1}{\lambda}(-2(1-\pi_0)\pi_1(1-\pi_1)) - \dfrac{2 \, \pi_1^2(1-\pi_1)^2(1-\pi_0)}{\lambda^2 + \lambda \left[ \lambda_p^2 \, \pi_0(1-\pi_0) + \pi_1(1-\pi_1) \right]} \end{bmatrix} \tag{64}$$

## A.6   Additional Results

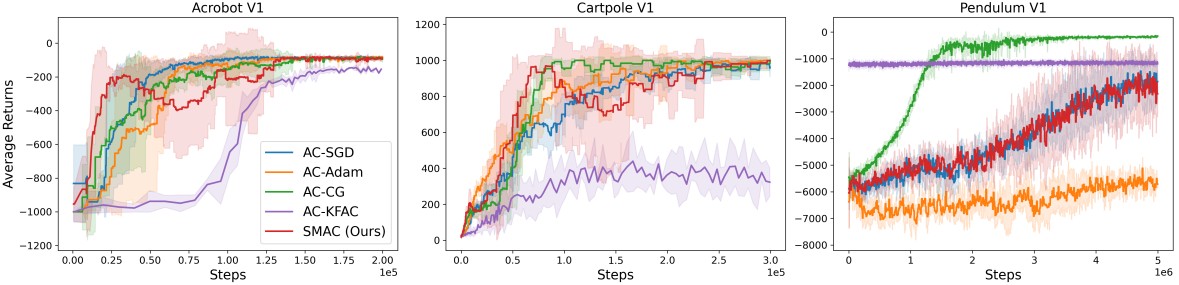

Figure 8: Raw results on three classic control tasks. These correspond to the results reported in Fig. 6, but without binning or smoothing.

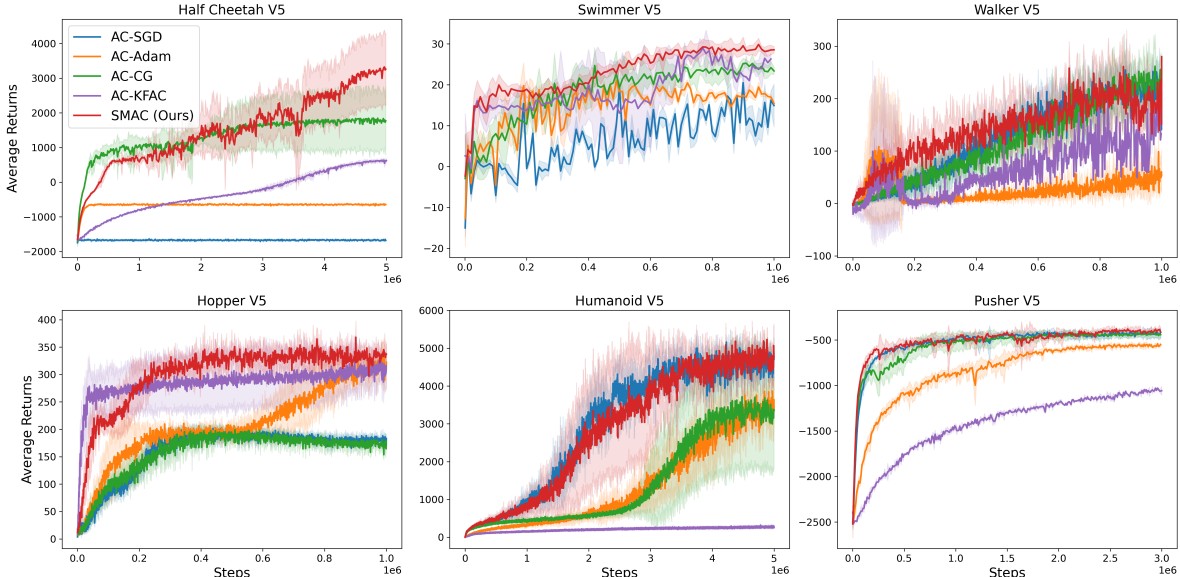

Figure 9: Results on six MuJoCo tasks. These correspond to the results reported in Fig. 7, but without binning or smoothing.

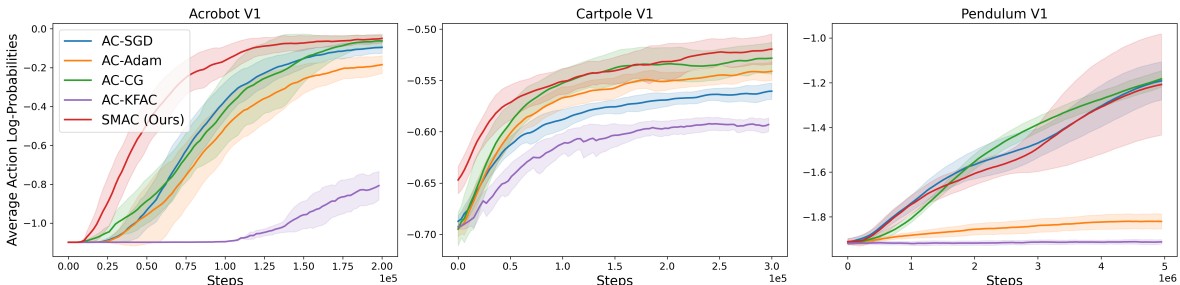

Figure 10: Average log-probabilities of the actions taken on three classic control tasks. All results are averaged over 5 random seeds, with shaded areas showing standard deviation.

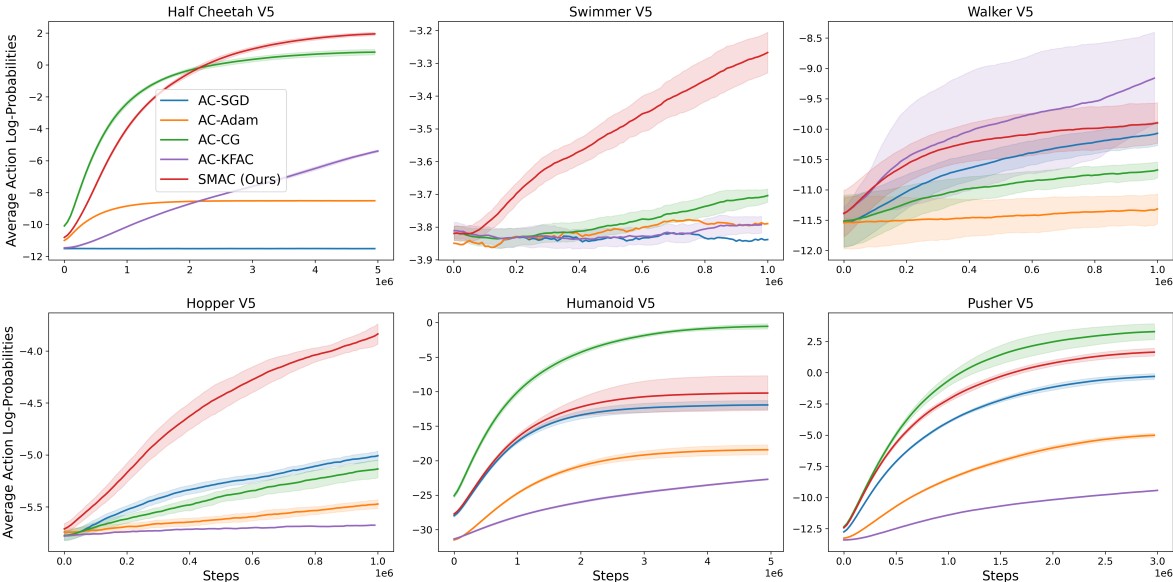

Figure 11: Average log-probabilities of the actions taken on six MuJoCo tasks. All results are averaged over 5 random seeds, with shaded areas showing standard deviation.

## A.7 Hyper-parameters

We consider the following hyper-parameters:

- $\eta$ (actor): step size for the policy-network update.

- $\alpha$ (critic): learning rate for the value network, optimized with Adam.

- $T$: number of environment steps collected before each parameter update.

- $\gamma$: reward discount factor.

- $\lambda_{\text{GAE}}$: trace-decay parameter used by Generalized Advantage Estimation when computing the advantage values $A_t$.

- $\lambda$: damping factor used in the matrix-free Sherman-Morrison approximation of the empirical Fisher.

We tune five of these on each environment. We omit tuning $T$, as preliminary runs have shown that performance is not highly correlated with its value. We select values for $\alpha$ (critic) from the interval $[1 \times$

| Algorithm | $\eta$ (actor) | $\lambda$ |
|---|---|---|
| SMAC | $\left[5\times10^{-5}, 5\times10^{-1}\right]$ | $\{10^{-4}, 10^{-3}, 10^{-2}, 10^{-1}, 1\}$ |
| AC-Adam | $\left[5\times10^{-6}, 1\times10^{-2}\right]$ | - |
| AC-SGD | $\left[5\times10^{-5}, 5\times10^{-1}\right]$ | - |
| AC-CG | $\left[5\times10^{-3}, 9\times10^{-1}\right]$ | - |
| AC-KFAC | $\left[1\times10^{-5}, 1\times10^{-3}\right]$ | - |

Table 2: Hyper-parameter values evaluated during tuning. Additionally, for each algorithm, we select values for $\alpha$ (critic) from $[1\times10^{-5}, 5\times10^{-3}]$, for $\gamma$ from $\{0.9, 0.95, 0.99\}$, and for $\lambda_{\text{GAE}}$ from $\{0.8, 0.85, 0.9, 0.95, 0.99\}$. See Table 5 for additional details on tuning hyperparameters specific to AC-KFAC.

| Environment | Algorithm | $\eta$ (actor) | $\alpha$ (critic) | $T$ | $\gamma$ | $\lambda_{\text{GAE}}$ | $\lambda$ |
|---|---|---|---|---|---|---|---|
| Acrobot | SMAC | $5\times10^{-2}$ | $1\times10^{-3}$ | 1000 | 0.99 | 0.9 | 0.1 |
| | AC–Adam | $6\times10^{-4}$ | $1\times10^{-3}$ | 1000 | 0.99 | 0.9 | - |
| | AC–SGD | $2\times10^{-1}$ | $1\times10^{-3}$ | 1000 | 0.99 | 0.9 | - |
| | AC–CG | $6\times10^{-1}$ | $1\times10^{-3}$ | 1000 | 0.99 | 0.9 | - |
| | AC-KFAC | $1\times10^{-2}$ | $1\times10^{-3}$ | 1000 | 0.99 | 0.9 | - |
| Cartpole | SMAC | $5\times10^{-3}$ | $1\times10^{-3}$ | 1000 | 0.99 | 0.9 | 0.1 |
| | AC–Adam | $7\times10^{-5}$ | $1\times10^{-3}$ | 1000 | 0.99 | 0.9 | - |
| | AC–SGD | $7\times10^{-3}$ | $1\times10^{-3}$ | 1000 | 0.99 | 0.9 | - |
| | AC–CG | $8\times10^{-2}$ | $1\times10^{-3}$ | 1000 | 0.99 | 0.9 | - |
| | AC-KFAC | $1\times10^{-3}$ | $1\times10^{-3}$ | 1000 | 0.99 | 0.9 | - |
| Pendulum | SMAC | $6\times10^{-3}$ | $1\times10^{-3}$ | 1000 | 0.99 | 0.9 | 0.1 |
| | AC–Adam | $7\times10^{-4}$ | $1\times10^{-3}$ | 1000 | 0.99 | 0.9 | - |
| | AC–SGD | $5\times10^{-2}$ | $1\times10^{-3}$ | 1000 | 0.99 | 0.9 | - |
| | AC–CG | $3\times10^{-2}$ | $1\times10^{-3}$ | 1000 | 0.99 | 0.9 | - |
| | AC-KFAC | | | | | | |

Table 3: Training hyper-parameters for the classic control environments.

$10^{-5}, 5\times10^{-3}]$, for $\gamma$ from the set $\{0.9, 0.95, 0.99\}$, and for $\lambda_{\text{GAE}}$ from the set $\{0.8, 0.85, 0.9, 0.95, 0.99\}$. For the rest of the hyper-parameters, we select values according to Tables 2 and 5. The goal is to find hyper-parameter configurations that maximize the return averaged over the last 100 epochs, for each environment and algorithm.

We find that learning rates have the highest impact on performance. On the contrary, $\gamma = 0.99$ and $\lambda_{\text{GAE}} = 0.9$ perform well in all environments. For each environment and algorithm, we select the hyper-parameter configuration that performs best, and then use it to generate the results in Sec. 6. These configurations are shown in Tables 3 and 4 (and Table 5 for AC-KFAC). For hyper-parameters specific to AC-KFAC, we found a damping of $1 \times 10^{-3}$, factor decay of 0.99, KL clipping value of $1 \times 10^{-3}$, and an update frequency of 1 iteration to perform well in all environments except the ones specified in Table 5.

| Environment | Algorithm | $\eta$ (actor) | $\alpha$ (critic) | $T$ | $\gamma$ | $\lambda_{\text{GAE}}$ | $\lambda$ |
|---|---|---|---|---|---|---|---|
| Half Cheetah | SMAC | $8\times10^{-4}$ | $6\times10^{-4}$ | 1000 | 0.99 | 0.9 | 0.01 |
| | AC–Adam | $5\times10^{-3}$ | $6\times10^{-4}$ | 1000 | 0.99 | 0.9 | - |
| | AC–SGD | $1\times10^{-1}$ | $9\times10^{-4}$ | 1000 | 0.99 | 0.9 | - |
| | AC–CG | $9\times10^{-1}$ | $8\times10^{-4}$ | 1000 | 0.99 | 0.9 | - |
| | AC-KFAC | $1\times10^{-3}$ | $1\times10^{-3}$ | 1000 | 0.99 | 0.9 | - |
| Hopper | SMAC | $1\times10^{-2}$ | $1\times10^{-4}$ | 1000 | 0.99 | 0.9 | 1.0 |
| | AC–Adam | $1\times10^{-4}$ | $1\times10^{-4}$ | 1000 | 0.99 | 0.9 | - |
| | AC–SGD | $1\times10^{-3}$ | $1\times10^{-4}$ | 1000 | 0.99 | 0.9 | - |
| | AC–CG | $1\times10^{-2}$ | $1\times10^{-4}$ | 1000 | 0.99 | 0.9 | - |
| | AC-KFAC | | | | | | |
| Swimmer | SMAC | $3\times10^{-2}$ | $1\times10^{-4}$ | 1000 | 0.99 | 0.9 | 1.0 |
| | AC–Adam | $1\times10^{-4}$ | $1\times10^{-4}$ | 1000 | 0.99 | 0.9 | - |
| | AC–SGD | $1\times10^{-3}$ | $1\times10^{-4}$ | 1000 | 0.99 | 0.9 | - |
| | AC–CG | $1\times10^{-2}$ | $1\times10^{-4}$ | 1000 | 0.99 | 0.9 | - |
| | AC-KFAC | $1\times10^{-2}$ | $1\times10^{-4}$ | 1000 | 0.99 | 0.9 | - |
| Walker | SMAC | $3\times10^{-2}$ | $1\times10^{-5}$ | 1000 | 0.99 | 0.9 | 1.0 |
| | AC–Adam | $1\times10^{-5}$ | $1\times10^{-5}$ | 1000 | 0.99 | 0.9 | - |
| | AC–SGD | $1\times10^{-4}$ | $1\times10^{-5}$ | 1000 | 0.99 | 0.9 | - |
| | AC–CG | $1\times10^{-2}$ | $1\times10^{-5}$ | 1000 | 0.99 | 0.9 | - |
| | AC-KFAC | $1\times10^{-3}$ | $1\times10^{-4}$ | 1000 | 0.99 | 0.9 | - |
| Humanoid | SMAC | $1\times10^{-4}$ | $1\times10^{-3}$ | 1000 | 0.99 | 0.9 | 0.01 |
| | AC–Adam | $3\times10^{-4}$ | $1\times10^{-3}$ | 1000 | 0.99 | 0.9 | - |
| | AC–SGD | $1\times10^{-3}$ | $1\times10^{-3}$ | 1000 | 0.99 | 0.9 | - |
| | AC–CG | $9\times10^{-1}$ | $1\times10^{-3}$ | 1000 | 0.99 | 0.9 | - |
| | AC-KFAC | | | | | | |
| Pusher | SMAC | $1.5\times10^{-3}$ | $1\times10^{-3}$ | 1000 | 0.99 | 0.9 | 0.01 |
| | AC–Adam | $2.5\times10^{-3}$ | $1\times10^{-3}$ | 1000 | 0.99 | 0.9 | - |
| | AC–SGD | $1\times10^{-1}$ | $1\times10^{-3}$ | 1000 | 0.99 | 0.9 | - |
| | AC–CG | $9\times10^{-1}$ | $1\times10^{-3}$ | 1000 | 0.99 | 0.9 | - |
| | AC-KFAC | $1\times10^{-2}$ | $1\times10^{-3}$ | 1000 | 0.99 | 0.9 | - |

Table 4: Training hyper-parameters for the MuJoCo environments.

| Hyper-parameter | Tuning values | Environment | Final value |
|---|---|---|---|
| Damping | $\left[5 \times 10^{-5}, 5 \times 10^{-2}\right]$ | Pendulum | $1 \times 10^{-2}$ |
| | | Humanoid | $1.5 \times 10^{-2}$ |
| Decay factor | $\{0.85, 0.9, 0.95, 0.99, 1.0\}$ | Pendulum | $0.85$ |
| | | Humanoid | $0.9$ |
| KL clip | $\left[5 \times 10^{-5}, 3 \times 10^{-2}\right]$ | Pendulum | $2.5 \times 10^{-2}$ |
| | | Humanoid | $4 \times 10^{-3}$ |
| Update frequency | $\{1, 2, 4, 8, 16, 32, 64\}$ | Pendulum | $16$ |
| | | Humanoid | $64$ |

Table 5: Hyper-parameter values evaluated when tuning AC-KFAC on each environment. We include the values we select for the Pendulum and Humanoid environments. All other environments use $1 \times 10^{-3}$ damping, 0.99 factor decay, $1 \times 10^{-3}$ KL clip, and 1 update frequency. These tuning values and final values complement Tables 2, 3, and 4.

