# OpenReview forum: "Rank-1 Approximation of Inverse Fisher for Natural Policy Gradients in Deep Reinforcement Learning"
_TMLR — Accepted by TMLR_

### Review · Reviewer_Hkmz · 2025-09-15

**Summary Of Contributions:**

This paper studies a rank 1 monte carlo approximation of the inverse fisher, in an attempt to follow natural gradients in policy approximation for deep RL. The main idea is to use the Sherman-Morrison formula to calculate the inverse of required for the inverse fisher, which can be done because the rank 1 monte carlo approximation is simply a cross product of the gradients of the log policy. This is, of course, much more efficient than directly calculating the matrix inverse. The core of the paper is a theorem showing convergence of the method, under some basic assumptions, along with experimental validation.

## Strengths
In my view, the strengths of the paper are threefold. First, it is well written and organized—by and large it is a pleasure to read. Second, the idea of using the Sherman Morrison formula to approximate the inverse fisher is an elegant idea—I’m almost surprised it hadn’t been done yet! Third, it is great to have both theoretical verification of convergence for the method and the empirical work together, making a much stronger contribution than either of the two alone.

## Weaknesses
1. There are some choices in the paper that seem to be poorly justified given their importance
2. There are some issues in the math and proof that, while I expect they do not invalidate the proof, make it unnecessarily difficult to parse and thus need to be addressed. For example, parts of the proof are poorly explained, there seem to be (minor) errors in a number of places, and, most notably, there are quantities and certain uses of notation that are not properly defined
3. I am concerned about the measures of variance that are being plotted in the figures. What are the error bars? Is the data smoothed before calculating them?

**Additional Comments:**

Note that my knowledge of current research is more focused on deep learning than RL, so I cannot easily comment on how the current paper fits into the broader literature. Please take this into account when considering my review.

**Audience:**

Yes

**Audience Explanation:**

Given that the paper is about policy gradient estimation, if the requested changes section can be addressed then I think the paper could be of interest to large subsets of the RL community

**Broader Impact Concerns:**

Given the increasing use of RL methods for not only societally beneficial but also damaging applications, it would be nice to discuss some of the negative applications and what might need to be done to mitigate them

**Claims And Evidence:**

No

**Claims Explanation:**

There are some issues around exposition in the proof, and, potentially, around the standard deviations in the plots that require attention before I can answer yes to this question. See sections "weakness 2" and "weakness 3" in the requested changes section for concrete examples.

**Requested Changes:**

I have organized the requested changes into sections based on the three main weaknesses and then unrelated edits (‘misc edits’). I have added “**C**” at the end of changes that I view as critical for my review, and “**S**” after ones that I view more as suggestions.

## Weakness 1
- Section 4.1 (last equation page 4, and text above it): could you explain why you use the identity and damping coefficient instead of using the previous estimate of the FIM here, or simply setting $\lambda = 0$? Some justification would be great. **C**
- Section 4.1 (last equation page 4, and text above it): it would be nice to include some study of how the algorithm works when the previous estimate is reused, and how varying $\lambda$ affects learning. **S**
- Figure 1: it would be nice to discuss a little bit why the batched version performs worse than the original version. I assume because of bias introduced through the averaging of the vector before taking the cross product? **S**
- Second equation section 5.1: $\bar{l}^k$ is a sample mean, and thus the “approximation” being made is that the expected value of the cross product of $l^k$ with itself can be written as the cross product of the expected values. Presumably this would require that $l^k$ is statistically independent of itself which is of course not the case. Could the authors provide a sentence or two mentioning this assumption, providing some justification, and then, at least in brief, explaining the bias introduced? **C**
- Last paragraph of results section: what is the justification for plotting log probabilities? Is this done in the literature? It would be good to provide some justification and, if precedent exists, some references.
How were hyperparameters selected?

## Weakness 2
- Equation 2: the expected value on the RHS is over the trajectory, which should make the expectation not depend on state, but the LHS writes $F_s$ as a function of $s$. Why is this? Presumably $s$ is the initial state? If so, should this expression include a sum over time? **C**
- First equation on page 4: the sum is just a single sum over trajectories, but shouldn’t this be a double sum over trajectories, $i$, and time within a given trajectory, $t$, with $a$ and $s$ depending on both? **C**
- First equation in section 4.1 (on the top line of the equation RHS): I think $\hat{F}^{k-1}$ should be $(\hat{F}^{k-1})^{-1}$ **C**
- Section 4.1 (last equation page 4): $\lambda$ is not defined as a damping coefficient until rather later. Should ideally define right after the equation **S**
- First equation page 5: if the superscript $k$ denotes the learning iteration then it would make sense to index the $\theta$ values in the RHS of this equation with $k$. Also, it is weird to write $F^k(\theta)$ on the RHS if $F$ depends on $k$ only through $\theta$, and also it is inconsistent with the rest of this part of the paper to write $F$ explicitly as a function of $\theta$ **S**
- First equation page 5: from which point in the trajectory are $a$ and $s$ sampled in this equation? **C**
- After Equation 5 and in Algorithm 1: the advantage function is included in the gradient after equation 5 but it is missing in line 7 of Algorithm 1.
- Theorem 1 statement: It would be good to add the assumptions mentioned above Theorem 1 into the italic part of the theorem statement or, at the very least, organize the assumptions cleanly in the preceding paragraph and then write “under the assumptions listed above…” in the Theorem statement **S**
- Theorem 1 statement: the variables $L_J$, $\varepsilon_{bias}$, and $\sigma^2$ are in the statement of Theorem 1 are not defined—please fix! **C**
- Theorem 1 statement: it would also be good to define $K$, $N$, and $H$ either right after or before the Theorem statement **S**
- Algorithm 1 line 10: missing time indices on $R^k$ and value function.
- First equation in section 5.1: if I am not mistaken, $a$ and $s$ should be indexed by both $i$ and $t$, and there should be a double sum over the indices, right? **C**
- Last equation on page 7: $\Delta\theta^k$ is written $w^k$ in the proof (in the appendix), right? It would be great to make the notation consistent **C**
- Assumption 3: could you provide more context about this assumption? When is it true? When does it not hold? Can you reference other works that use the assumption and/or study it? What does $L$ represent intuitively? **C**
- Equation 11: is the first equality in this equation a definition? If so, perhaps use $\equiv$. **S**
- Equation 11: what is $\nu$? Please define **C**
- Helper Lemmas: the following are not defined: $L_J$ (note that it might be better to choose a different letter given the use of $L$ in assumption 3), $J^H$, $R$, $\sigma$. Please define these. **C**
- Lemmas: please reference specific proposition/lemma/theorem numbers in the original papers for each of the referenced lemmas **S**
- Lemma 1: I don't think this is used in the proof? If so, please remove **C**.
- Lemma 3: why is $g^k$ defined here and then not used in the equation? Perhaps this should be $\nabla J^H$? If so, please correct throughout the paper to make notation consistent **C**
- Lemmas 2, 3: I could not find these results in the referenced papers. Could you point them out to me? Thanks **C**
- Lemma 4: it seems that the notation $F_\phi$ is used here but not in the following section (the subscript $\phi$ is dropped). Please make consistent **C**
- Section A.3: it is written: “we know from Jensen’s inequality and [EQUATION]”, but the equation just shows a particular case of Jensen’s inequality. This redundancy can be removed, or replaced with something else that was used in the derivation **C**
- Line 4 of Equation 14: missing $||^2$ on RHS **C**
- Top of page 16: I’m not sure that it is actually Lemma 3 and Assumption 2 that are used here. Maybe Lemma 2 and remark 2?? **C**
- Equation 15 \sigma^2 not defined **C**
- Equation 15: it seems that the inequality $(x+y)^2 \leq 2x^2 + 2y^2$ was used twice here; perhaps there is a factor of $2$ missing somewhere? **S**
- Second last equation on page 16: $H_{,\star}$ should be $H_{\star}$ I think **S**
- Last equation on page 16: missing $16L_J$ on RHS **C**
- $\sigma^2$ is treated as a constant but I’m a little concerned that it might depend on $H$, and potentially $N$. Could you provide some insight here? **C**
- Page 18: it is written: “In summary, we require $N$ and $K$ to satisfy equation 18, equation 20, and equation 22, which leads to…”. I think I would also reference $H$ having to satisfy Equation 18 in this sentence **S**
- Page 18, second last equation: would you please show the derivations for the conditions on $H$, $N$, and $K$, in the second last equation? **C**

## Weakness 3
- Figures: are the shaded regions in the figures the true standard deviations of the learning trajectories, or are they the standard deviations after applying the binning and smoothing protocol mentioned right before section 5.1? If the latter case is true, it would be nice to plot the unsmoothed, unbinned data in the supplementary. Also, it would be great to explain this information about visualizing variance in the figure caption. **C**

## Misc. Edits
- 5th line after Equation 3: I don’t entirely think the sentence needs to start with “Specifically, …”. I believe this could be deleted for more concise writing. **S**
- Table 1: I assume what is displayed are means $\pm$ standard deviations, but it would be great to make this explicit in the table caption **C**
- On page 7 it is mentioned: “To better handle continuous control problems, SMAC and all the baselines we compare against compute advantages using the GAE”. Could you elaborate on this please?
- 5.1 title (and again later in the appendix): why is the effect of batch side labeled as an “ablation”? This seems like a weird wording choice—perhaps reword? **S**
- In the last paragraph of the results section it is written: “Finally, we turn our attention to the average log-probabilities of the actions taken during training”, but no figure is referenced (I believe the figure being discussed is in the appendix?). It would be good to reference the correct figure here.
- Equation 8: it could be nice to define the matrix inequality used here **S**

---

> ### Author Response · Authors · 2025-11-06
>
> We thank you for your detailed feedback, corrections, and ideas. We believe we successfully addressed your concerns.
>
> Weakness 1:
>
> For 1,2,
> We added discussions and experiments for the previous EF estimate used vs. our method, and for different damping coefficients.
>
> For 3,4,
> We introduced the bias that arises from approximating the expected value of the cross product by the cross product of the expected values. And this bias causes the performance difference between the batch version and the single-sample version
>
> For 5,
> We provided better justifications for plotting log probabilities, bringing attention to our process for measuring agents’ confidence in their actions and identifying overconfident sub-optimal agents.
>
>
> Weakness 2:
>
> For 1,
> We have cleared out the definition in the paper and would like to point out that FIM for policy $\pi_\theta(\cdot|s)$ can be written over the states sampled from the initial state distribution measure (which is equation 2) and then can be averaged over the state distribution, giving rise to the final FIM $F_{\rho}$.
>
> For 2.
> You have pointed out exactly what we have written. The outer expectation is over all the trajectories sampled via policy $\pi_\theta$ and the inside summation is over the time dimension, making the expression equivalent to a double summation.
>
> For 3.
> Our expression has an inverse exponent (-1) on top. To make it clearer, we have changed the curly brackets to square brackets.
>
> For 5,
> We have added a note in Section 3 before the paragraph [Sherman-Morrison for computing the inverse Fisher] to convey that, this notation has been adopted for simplification throughout the sections and proofs.
>
> For 6,
> One-sample empirical fisher and $s$, $a$ sampled over the trajectory.
>
> For 10.
> The variables $K$, $N$ and $H$ have been defined in the discussion part following the Theorem 1 in our paper.
>
> For 12.
> Thank you for pointing this out. In our setup, B denotes the total number of state-action transitions across trajectories, so each $a_i$, $c_i$, represents one transition. We have clarified this in the text to avoid confusion.
>
> For 13,
> They are similar, but to be exact $\Delta \theta^k = \eta w^k$
>
> For 15,
> This definition and the constant stems from very early works of Agrawal et al 2020 and have been used extensively in theoretical papers related to policy gradients, so we have chosen to keep it that way.
>
> For 20,
> We would like to clarify that $\nabla J^H$ is the true gradient direction for the objective function where the trajectories are truncated to horizon H, whereas $g^k$ signifies the batch average (N samples) of H-horizon trajectories sampled from the policy.
>
> For 21,
> Lemma B.1 of Liu et al. 2020 has Lemma 2, and Equation F.2 of Liu et al. 2020 provides an upper bound to the expectation of the H-horizon policy gradient direction.
>
> For 27,
> The factor of 2 is in each term of the last part of the inequality in Equation 15 of the previous revision.
>
> For 28,
> To keep the notations consistent with past papers, we have kept the notation as $J^{H, \star}$ throughout the proofs.
>
> For 30,
> Thank you for pointing that out. $\sigma^2$ is the smooth uper bound of the variance of a gradient from single trajectory truncated until horizon H. So its is implied that it is not dependent on N, but might depend on horizon H. As the horizon H is finite (of the order $\mathcal{O}(\log((1-\gamma)^{-1}\varepsilon^{-1}))$) so we can consider a smooth upper bound which is finite for parametric gaussian policies. We have added this as a new remark in Appendix A.2.
>
> For 4, 7, 9, 11, 14, 16, 19, 22, 23, 24, 25, 26, 29, 31, 32,
> Thank you for pointing these out. We have added the changes in our new revision.
>
> Weakness 3:
> We also discussed our hyperparameter tuning process in the Appendix.
>
> We explained, in both text and captions, how we compute the standard deviation after binning and smoothing. We additionally added new plots showing the raw data in the appendix, and referred to them in the main text.
>
> We also addressed all the minor concerns, improving captions, explanations, and justifications, adding missing references, and defining missing terms.
>
> We hope our changes are satisfactory and look forward to your response.

---

> > ### Comment · Reviewer_Hkmz · 2025-11-15
> > **Reviewer Follow-up**
> >
> > Thank you very much for the updates! I’m sorry about sending this with limited time to spare but I still have some reservations regarding several points, mostly related to “weakness 2”. Note that the numbers correspond to the ordering of bullet points in my original review.
> >
> > **Weakness 2**
> > - 2: in the first equation on page 2 there is only one sum, as far as I can see, where $a_i$ and $s_i$ are indexed according to the trajectory, $\tau_i$, index. Is this to suggest that there is only one state/action pair, $(a_i, s_i)$, taken from each trajectory?
> > - 6: related to the above point, to clarify, from section “Sherman-Morrison for computing the inverse Fisher” onwards (until the batch approximation) only 1 sample monte carlo was used to estimate $\hat{F}$?
> > - 19: Lemma 1: I am still confused about where Lemma 1 is used in the proof. In particular, Lemma 1 and Lemma 3 bound the same quantity, and I see where Lemma 3 was used but not Lemma 1
> > - 20: my question here is not about the difference between $\nabla J^H$ and $g^k$, but why $g^k$ is defined in the statement of Lemma 3 and then not used. Why not define it somewhere else?
> > - 21: In your response you mention that Lemma 2 is Equation F.2 in Liu et al. but you still cite Xu et al. Importantly, I have checked both of these papers and cannot find an Equation F.2 in either of them.
> > - 22: you mentioned that you resolved the notational issue that I mentioned but it appears to still be there
> > - 27: to clarify, I see that you have included a factor of $2$ but, if I am not mistaken, you used the inequality that introduces a factor of $2$ twice, therefore I would expect a factor of $4$.
> > - 29: thank you for adding the factor of $16$ that was missing, but are you not still missing $L_J$ too?
> > - 30: I am still a bit concerned about the justification you have used for why the $H$ dependence of $\sigma$ can be ignored. In particular, you mention that you consider finite horizons but the RL problem that you study, as introduced in the main body of the paper, is technically an infinite horizon problem if I am not mistaken. Moreover, other portions of your proof still rely on taking $H$ to be sufficiently large. Lastly, you talk about the importance of smooth parametric policies for this particular assumption but you do not justify this here or reference any research that does.
> > - 32: I was hoping for a little more detail on the derivation of $H$, $N$, and $K$, in the second last equation of section A.4, and this was not provided.
> >
> > **Broader Impact**
> > - Broader impact concerns: I mentioned some broader impact concerns that could be nice to engage with, at least at a high level. For example, RL is being used more frequently for fine-tuning of LLMs, and LLMs are having significant negative impacts on society (via copyright infringement during training, environmental impacts, propagation of misinformation, etc). Alternatively, RL provides an interesting framework for power grid optimization, that could provide energy savings and thus aid in climate goals. I imagine that reducing variance through natural gradients in RL could thus, ultimately, have both positive and negative societal impacts. These could be useful to mention, at least briefly.
> >
> > I would ideally like to see all of these issues addressed before I could recommend this paper for acceptance. However, I also wish to highlight that points 21, 30, 32 are of particular importance to me.
> >
> > Lastly, thank you for addressing my concern around the variance in the plots. I was very happy to see this change in the revision!

---

> > > ### Author Response · Authors · 2025-11-21
> > >
> > > Thank you for your detailed and constructive feedback. We have revised the manuscript accordingly. Below we address the numbered comments:
> > >
> > > 2. Thank you for pointing out this confusing notation.  In this equation we estimate the empirical Fisher from all state–action pairs within a single trajectory. The index i in $(s_i, a_i)$ was meant to enumerate time steps along that trajectory.
> > >
> > > 6. In this equation, $(s,a)$ is intended to be a single state–action pair sampled from the trajectory, we updated it in the new revision to make it more clearer.
> > >
> > > 19. You are right that the original Lemma 1 and Lemma 3 bound the same quantity. In the revised version we remove the old Lemma 1 and renumber the old Lemma 3 as Lemma 1.
> > > 20. We removed the $g^k$ in lemma 3 in the new revision
> > >
> > > 21. Thank you for pointing this out. You are correct that Lemma 2 in our paper should refer to Equation F.2 in the appendix of Liu et al. (2020), not Xu et al. We have corrected this citation in the new version. The reason you could not find Equation F.2 is that the link we provided in the references pointed to a version of Liu et al. that does not include the appendix. We have now updated the reference to a version that includes the full appendix, where Equation F.2 appears.
> > >
> > > 22. We deleted the subscript in the new revision.
> > >
> > > 27. Thank you for carefully checking the constants in this inequality, We have fixed this and the following related factors in the new revision.
> > >
> > > 29. $L_J$ is added in the new revision.
> > >
> > > 30. In the revised version, we have updated Remark 3 to clarify our variance assumptions and provide specific references. In particular, we now explicitly state that, in our analysis based on finite truncated horizons, we assume bounded variance for the policy gradient estimator under smooth parameterized (Gaussian) policies; see Zhao et al. (2012, Section 3) and Pirotta et al. (2013, Section 5.1) for details. We also note that while the variance $\sigma$ may in general depend on the horizon $H$ at most linearly, our proofs only require a uniform upper bound over the finite range of truncation horizons used in the analysis.
> > >
> > > 32. A brief derivation of H, N and K is added to the new revision.
> > >
> > > Also, we have added a brief Broader Impact paragraph to discuss the potential positive and negative societal implications. We hope these clarifications and revisions address the your concerns, and we thank you again for your careful reading of our paper.

---

### Review · Reviewer_D2Gk · 2025-09-18

**Summary Of Contributions:**

The paper presents an approach to obtain an approximation of the inverse Fisher information matrix with cost linear to the number of policy parameters. The technique is implemented in a policy gradient method and evaluated on several control tasks. In addition, a theoretical convergence analysis is provided.

**Audience:**

Yes

**Audience Explanation:**

To the best of my knowledge, the application of the Sherman-Morrison formula to obtain an approximation of the inverse Fisher information matrix is novel in the context of natural policy gradient method.

**Broader Impact Concerns:**

I think such discussion is not needed for this kind of paper.

**Claims And Evidence:**

No

**Claims Explanation:**

A theoretical analysis is provided, although some details are not clear to me due to presentation issues. For instance, the bound \epsilon_bias, defined in the second assumption (see appendix), hides terms that depend both on the transition function of the MDP and the expressivity of the critic approximator, which can be large and may not really be controllable in practice. For this reason, while the algorithm may converge, but its limit may be far from the optimal point. In addition, the analysis seems to largely rely on previous results (all the helper lemmas in the appendix). The authors should provide a more detailed discussion on those points.

I think some important information about the experiments is missing to fully appreciate the experimental evaluation:
- How were the hyperparameters chosen for all the methods?
- Could the authors explain in more details why the computation costs can be very high (2nd sentence in Section 5.1)?
- Could the authors provide some explanations or conjectures about which type of environments/tasks the proposed algorithm would be expected to work well or not?

**Requested Changes:**

A more precise discussion of the theoretical analysis should be provided. Otherwise, it feels completely disconnected from the empirical evaluation. Would it be possible to design some experiments, possibly in toy domains, to verify the obtained bounds?

The missing information about the experiments (see above) should be provided.

Although the exposition is generally quite clear, the paper in its current version is hard to read due to notational and formatting issues (see below). I suggest the authors to improve the exposition to make it more pedagogical. Currently, some explanation or definition are not provided in the main paper, and the reader needs to search for them (for example in the appendix).

Notation issues:
	The MDP model is introduced as episodic, but equations are provided in the infinite horizon case.
	All notations should be explained the first time they appear.
	The subscript in the definition of the value function V and Q are undefined and not needed.
	The expectation over episodes in (2) should be over states.
	The terms in the equation for the empirical Fisher information don't have any subscript i.

Formatting issues:
The citations that are not part of a sentence should be changed from "Authors (year)" to "(Authors, year)"

---

> ### Author Response · Authors · 2025-11-06
>
> Thank you for taking the time to provide feedback on our work. We believe our changes have addressed your concerns.
>
> We have addressed this concern in the paragraphs before and after Theorem 1 in the new revision.
>
> We gave a detailed explanation of our approach for selecting hyperparameters in the Appendix.
>
> We gave an explanation about the computational cost of one-sample update in practice.
>
> A toy setup will definitely help explain the difference between PG, exact NPG and our proposed method (SM-ActorCritic). For this, we have chosen the classical toy problem from [Bagnell et al. 2003], and we have included a small subsection to illustrate the difference between each update direction. Briefly speaking, we observe that for a diverse number of policy parametrizations, our method captures the direction as well as the magnitude of the update considerably better than standard PG estimate.
>
> We fixed the notation and defined terms properly, in the main text, as advised. We also fixed the errors found in our equations. To make the paper easier to follow and our contribution easier to isolate, we moved some of the methods found in the literature for computing the empirical Fisher to the Preliminaries section. The terms used in both these methods and ours are now defined properly. Finally, we fixed the formatting issues for our citations.
>
> We hope our modifications have addressed most or all of your main concerns. We are looking forward to your response.

---

### Review · Reviewer_snXP · 2025-10-18

**Summary Of Contributions:**

The authors study natural gradient policies (NPG) in reinforcement learning (RL). Computing the inverse Fisher Information Matrix (FIM) is a major bottleneck in natural gradient methods, particularly in deep RL. While various approximations exist (e.g., diagonal, K-FAC, Hessian-free), the authors propose a rank-1 approximation based on the Sherman–Morrison formula, which reduces computational complexity to $O(d)$, where $d$ is the number of model parameters. Under certain regularity assumptions on the policy, they establish a convergence guarantee (Theorem 1) that provides a sample-complexity bound of $O(\epsilon^{-4})$ for achieving an optimality gap less than the sum of two terms: one based on the approximation error of the advantage function and another for the desired precision level.

Empirically, the proposed SM-ActorCritic (SMAC) algorithm is compared to baseline methods using A2C with SGD, Adam, and conjugate gradient (CG) optimizers. Experiments on MuJoCo and classic control tasks show that SMAC achieves competitive or superior performance in several environments.

**Strengths**

- Fast, theoretically motivated rank-1 natural gradient approximation with $O(d)$ complexity.

- Convergence guarantee under regularity assumptions on the policy.

- Improved efficiency and competitive empirical performance across diverse RL benchmarks.

**Weaknesses**

- No direct comparison to other established natural-gradient approximations (e.g., K-FAC, empirical Fisher variants) in theory or experiments.

**Audience:**

Yes

**Audience Explanation:**

I think NPG methods remain important in RL for stability and potential sample efficiency. The method directly addresses the computational barrier of FIM inversion in deep models. Therefore, the paper might be interesting to some TMLR readers focused on optimization and theoretical RL.

**Broader Impact Concerns:**

To me, there are no major ethical concerns specific to this work. The paper focuses on algorithmic and theoretical advances in reinforcement learning optimization.

**Claims And Evidence:**

Yes

**Claims Explanation:**

The theoretical claims are consistent with prior analyses (e.g., Xu et al., 2020). Theorem 1 would benefit from explicitly restating key assumptions (e.g., Lipschitz smoothness and other assumptions in Xu et al., 2020) within the theorem statement. Empirical evidence aligns with the motivation, though broader baselines would strengthen the experimental part.

**Requested Changes:**

- Comparison with prior NPG approximations: Please expand empirical/theoretical comparisons with established methods (e.g., K-FAC, empirical Fisher variants, Hessian-free/TRPO). Moreover, please consider adding a comparison table of computational and theoretical properties and at least one experiment, including K-FAC or TRPO. This will better position SMAC compared to the existing approaches.

- Improving Theorem 1: Please include key assumptions (Lipschitz smoothness,...) within the theorem statement. Moreover, please report sample complexity in terms of the number of state–action pairs (not just trajectories). Since the horizon $H$ depends on $\epsilon$, multiplying by $H$ is needed.

- Numerical validation: In Table 1 (Half-Cheetah), other algorithms show large negative returns while SMAC is strongly positive. Please verify numbers.


- Your $O(\epsilon^{-4})$ bound is looser than the $\tilde{O}(\epsilon^{-3})$ bound reported for stochastic policy gradient (Yuan et al., 2021). Please discuss whether this gap stems from the rank-1 approximation, or proof technique/assumptions.

Rui Yuan, Robert M Gower, and Alessandro Lazaric. A general sample complexity analysis of vanilla policy gradient. arXiv preprint arXiv:2107.11433, 2021.

---

> ### Author Response · Authors · 2025-11-06
>
> Thank you for your feedback and for bringing attention to some of the issues in our work. We believe we successfully addressed your concerns.
>
> The AC-CG in our results uses TRPO’s NPG approximation (conjugate gradient with Fisher–vector products) but does not apply the trust-region KL constraint. Also, we added the KFAC results for comparison. KFAC generally achieves comparable or slightly better final returns than AC-CG, However, despite extensive tuning, KFAC performs noticeably worse in some environments.
>
> In our Theorem 1, we can see that the number of trajectories $N$ and the number of updates $K$ are of the order $\mathcal{O}(\frac{1}{\epsilon^2})$ but the number of trajectories required are in the logarithmic order, so we only present the leading terms. This simplification is not exactly accurate (as you have pointed out), but it clarifies the real dependence on the approximation error constant, the variance of policy gradient, and the discount factor. We would also like to point out that these notations have been previously adopted by Liu et al. Although, we think that this can be represented via $\tilde{\mathcal{O}}$ to make it more mathematically appropriate (like it has been done by [Yuan et al.] and we have added this change in our new revision.
>
> Thank you for pointing out the large negative returns in Half-Cheetah; this was an oversight on our side. We reran all algorithms and performed a more thorough hyperparameter tuning. We ensured that the low performance of standard A2C (either Adam or SGD) we achieved initially is correct, and not dependent on hyperparameters or an error in the implementation. Standard A2C is, therefore, simply not well-suited for the Half-Cheetah environment. For both SMAC and AC-CG, however, the tuning led to higher (positive) performance. Most importantly, the performance of AC-CG is now much closer to that of SMAC.
>
> Thank you for pointing out this paper’s results. We have added a small discussion part following our theorem to incorporate their findings. Briefly speaking, [Yuan et al.] uses the weak gradient domination assumption [Assumption 3.6], ABC assumption [Assumption 3.3] and expected Lipschitzness and smoothness of gradient assumption [Assumption 4.1] to find first-order stationary point convergence (FOSP) and arrive at $\mathcal{O}(\epsilon^{-3})$ and in [Liu et al. 2019] and our method, we find the sample complexity for average regret to the global optimum. In [Yuan et al. 2021] the catch is, this sample complexity only arises when the $\epsilon_{bias} = 0$ [see last paragraph of Section 4.3], while they also report (in the same paragraph) that for a particular value of $\epsilon`$ and $\mu$ (constants of ABC assumption), they recover back the same results from [Liu et al. 2019]. In that way, we would like to clarify that our bound is not “loose” per se, but stems from the choices of assumptions and proof strategy used.
>
> We hope our improvements addressed your main concerns, and we are look forward to hearing back from you.

---

> > ### Comment · Reviewer_snXP · 2025-11-17
> >
> > I thank the authors for the response. My main comments were addressed, and I do not have any further questions.

---

### Decision · Action_Editor_mJfz · 2026-01-10

**Recommendation:** Accept as is

**Additional Comments:**

I recommend the authors to triple check the new additions, specifically the new equations 36-40, toy problem sections, and the specific assumption about gradient variance bounds.

**Audience:**

Yes

**Audience Explanation:**

Yes, all reviewers are aligned that this is a good fit.

**Claims And Evidence:**

Yes

**Claims Explanation:**

Based on reviewer feedback and author responses, the claims made in the submission are now well supported. Initially, two of three reviewers answered "No," citing concerns about mathematical notation, proof justification, unclear bounds, and missing experimental details. The November 21st author revision successfully addressed these issues by adding KFAC comparisons, clarifying sample complexity bounds, improving theoretical discussion, and resolving key technical concerns about proof references and variance assumptions. The theoretical claims are now viewed as consistent with prior analyses with proper citations, while empirical evidence has been strengthened with additional comparisons and a toy example, leading to consensus that the evidence is adequate for acceptance.